# A deep learning approach to identify gene targets of a therapeutic for human splicing disorders

Dadi Gao [1,2,3,8], Elisabetta Morini [1,2,8], Monica Salani[1,8], Aram J. Krauson [1], Anil Chekuri [1,2], Neeraj Sharma[4], Ashok Ragavendran[1,3], Serkan Erdin [1,3], Emily M. Logan[1], Wencheng Li[5], Amal Dakka[5], Jana Narasimhan[5], Xin Zhao[5], Nikolai Naryshkin[5], Christopher R. Trotta[5], Kerstin A. Effenberger[5], Matthew G. Woll[5], Vijayalakshmi Gabbeta[5], Gary Karp[5], Yong Yu[5], Graham Johnson[6], William D. Paquette [7], Garry R. Cutting [4], Michael E. Talkowski[1,2,3✉] & Susan A. Slaugenhaupt [1,2✉]

Pre-mRNA splicing is a key controller of human gene expression. Disturbances in splicing due to mutation lead to dysregulated protein expression and contribute to a substantial fraction of human disease. Several classes of splicing modulator compounds (SMCs) have been recently identified and establish that pre-mRNA splicing represents a target for therapy. We describe herein the identification of BPN-15477, a SMC that restores correct splicing of *ELP1* exon 20. Using transcriptome sequencing from treated fibroblast cells and a machine learning approach, we identify BPN-15477 responsive sequence signatures. We then leverage this model to discover 155 human disease genes harboring ClinVar mutations predicted to alter pre-mRNA splicing as targets for BPN-15477. Splicing assays confirm successful correction of splicing defects caused by mutations in *CFTR*, *LIPA*, *MLH1* and *MAPT*. Subsequent validations in two disease-relevant cellular models demonstrate that BPN-15477 increases functional protein, confirming the clinical potential of our predictions.

[1] Center for Genomic Medicine, Massachusetts General Hospital Research Institute, Boston, MA, USA. [2] Department of Neurology, Massachusetts General Hospital Research Institute and Harvard Medical School, Boston, MA, USA. [3] Program in Medical and Population Genetics and Stanley Center for Psychiatric Research, Broad Institute of Harvard and MIT, Cambridge, MA, USA. [4] McKusick-Nathans Department of Genetic Medicine, Johns Hopkins University School of Medicine, Baltimore, MD, USA. [5] PTC Therapeutics, Inc., South Plainfield, NJ, USA. [6] NuPharmAdvise LLC, Sanbornton, NH, USA. [7] Albany Molecular Research Inc., Albany, NY, USA. [8] These authors contributed equally: Dadi Gao, Elisabetta Morini, Monica Salani. ✉email: MTALKOWSKI@mgh.harvard.edu; slaugenhaupt@mgh.harvard.edu

RNA splicing is a complex and tightly regulated process that removes introns from pre-mRNA transcripts to generate mature mRNA. Differential processing of pre-mRNA is one of the principal mechanisms generating diversity in different cell and tissue types. This process can give rise to functionally different proteins and can also generate mRNAs with different localization, stability, and efficiency of translation through alternative splicing of UTRs. RNA splicing requires the widely conserved spliceosome machinery along with multiple splicing factors[1]. The splicing reaction is directed by specific sequences, including the 5′ and 3′ splice sites, the intron branch point, and splice site enhancers and silencers found in both exons and introns[2]. Changes in the sequence of these elements, through inherited or sporadic mutations, can result in deficient or aberrant splice site recognition by the spliceosome. Disruption of splicing regulatory elements can generate aberrant transcripts through complete or partial exon skipping, intron inclusion or mis-regulation of alternative splicing, while mutations in the UTRs may affect transcript localization, stability, or efficiency of translation. Mutations that alter mRNA splicing are known to lead to many human monogenic diseases including spinal muscular atrophy (SMA), neurofibromatosis type 1 (NF1), cystic fibrosis (CF), familial dysautonomia (FD), Duchenne muscular dystrophy (DMD), and myotonic dystrophy (DM), as well as contribute to complex diseases such as cancer and diabetes[3–18]. The emergence of high throughput sequencing of large disease cohorts[19–21], and the remarkable efforts to aggregate and annotate these mutations in an accessible infrastructure such as ClinVar[22], now provides an unprecedented opportunity to apply deep learning approaches to predict mutations that affect pre-mRNA splicing[23]. The potential of developing such models will continue to increase as next-generation transcriptome sequencing (RNASeq) data are amassed and curation of the associated mutational processes matures[23–26].

Therapeutic approaches aimed at correction of pre-mRNA splicing defects, including antisense oligonucleotides, splicing modulator compounds (SMCs), and modified exon-specific U1 small nuclear RNA, have shown significant promise in many diseases[27–35]. SMCs are attractive because they can be optimized for broad tissue distribution and are orally administered[27,36,37]. With advances in precision medicine and the capability to discover patient-specific mutations, there is strong impetus to develop methods to predict if a drug might be beneficial in a specific patient. Deep learning techniques offer the potential to accomplish this at scale by integrating genomic data with annotation databases and relevant information about mutational mechanisms[38]. Deep learning models have been successfully applied to a spectrum of biological topics, including genotype-phenotype correlation studies[39], identification of disease biomarkers[40], and identification of protein binding motifs[41]. Here, we applied and optimized a specific deep convolutional neural network (CNN) to discover motifs that are likely to be responsive to BPN-15477, a potent SMC of the *ELP1* pre-mRNA carrying the major FD splice mutation IVS20 + 6T > C. We identified 155 genes harboring pathogenic ClinVar mutations, each predicted to disrupt pre-mRNA splicing, that could be corrected by BPN-15477 treatment, and validated several using minigenes or patient cells. These studies suggest that the integration of genomic information, clinical annotation of disease-associated variants, and deep learning techniques have significant potential to predict therapeutic targeting for precision medicine.

## Results

### Discovery of the splicing modulator BPN-15477. Several small-molecules have been developed to selectively modulate the

splicing of specific pre-mRNAs, offering potential treatments for SMA and FD[27,36,37,42–45]. One such compound, kinetin (6-furfurylaminopurine), was previously shown to promote exon 20 inclusion in the Elongator complex protein 1 gene (ELP1, MIM: 603722) in FD[44,46]. Although kinetin is a naturally occurring compound with a safe absorption, distribution, metabolism, and excretion (ADME) profile, very high doses are necessary to achieve modest *ELP1* splicing correction in vivo[47,48]. As part of the NIH Blueprint Neurotherapeutics Network, we identified a class of highly potent SMCs that selectively modulate *ELP1* pre-mRNA splicing and increase the inclusion of exon 20[49]. BPN-15477 (Fig. 1a) increases full-length *ELP1* mRNA by increasing exon 20 inclusion and is significantly more potent and efficacious than kinetin in our luciferase splicing assay (Fig. 1b,c) and in FD patient cell lines (Fig. 1d). Previously, we showed that a modest increase in ELP1 protein rescued neurologic phenotypes in a phenotypic mouse model of FD[50]. Therefore, we evaluated BPN-15477 in vivo to confirm that splicing correction can lead to a concomitant increase in ELP1 protein. We treated the *TgFD9* transgenic mouse[51], which carries the human *ELP1* gene with the major FD splice mutation, once daily via oral gavage for 7 days, and mice were sacrificed 1 h after the last dose. Treatment increased full-length *ELP1* transcript in a dose-dependent manner and, importantly, led to at least a two-fold increase in functional ELP1 protein in brain, liver, kidney, heart, and skin (Fig. 1e, f, Supplementary Fig. 1a–c). In addition, the treatment was well tolerated, no weight loss or adverse effects were observed in the treated groups, and the level of splicing correction correlated with BPN-15477 tissue distribution (Supplementary Fig. 1d–e). These results clearly demonstrate that treatment with BPN-15477, which corrects splicing of the *ELP1* transcript, significantly increases the level of functional protein in vivo in all tissues, including brain.

**Evaluation of BPN-15477 on transcriptome splicing**. To estimate the potential for BPN-15477 to influence transcriptome-wide splicing, we treated six wild-type (WT) human fibroblast cell lines with 30 μM BPN-15477 or vehicle (DMSO) for 7 days and performed RNA-seq to evaluate changes in exon inclusion (Supplementary Table 1). Splicing differences were determined by counting the RNASeq reads covering two junctions of three consecutive exons (exon triplets) and by comparing the change in percent spliced-in (ΔPSI or Δψ) of the middle exon after treatment (Fig. 2a, see "Methods")[52,53]. We identified 934 exon triplets that showed differential middle exon inclusion or exclusion in response to BPN-15477 treatment: 254 with increased exon inclusion (Δψ ≥ 0.1 and FDR < 0.1), and 680 with increased exon exclusion (Δψ ≤ −0.1 and FDR < 0.1, Fig. 2b and Supplementary Data 1). BPN-15477 modulates splicing selectively as we observed splicing changes in only 0.58% of all expressed triplets (934 out of 161,097 expressed triplets). To experimentally confirm the accuracy of the PSI changes measured by RNASeq, we performed independent treatment experiments and evaluated exon inclusion using RT-PCR (Fig. 2d–g, Supplementary Table 2) and found the estimated percent exon inclusion to be remarkably consistent with the calculated Δψ values (Fig. 2c).

**Convolutional neural network identifies sequence signatures responsible for BPN-15477 response**. Pre-mRNA splicing is regulated by both exonic and intronic sequence elements. These sequences govern interaction with the spliceosome and splicing factors and regulate the fate of exon recognition and inclusion[2,54]. We hypothesized that sequence signatures within the exon triplets are a key determinant of drug responsiveness. To identify the sequence motifs, we trained a CNN model using the

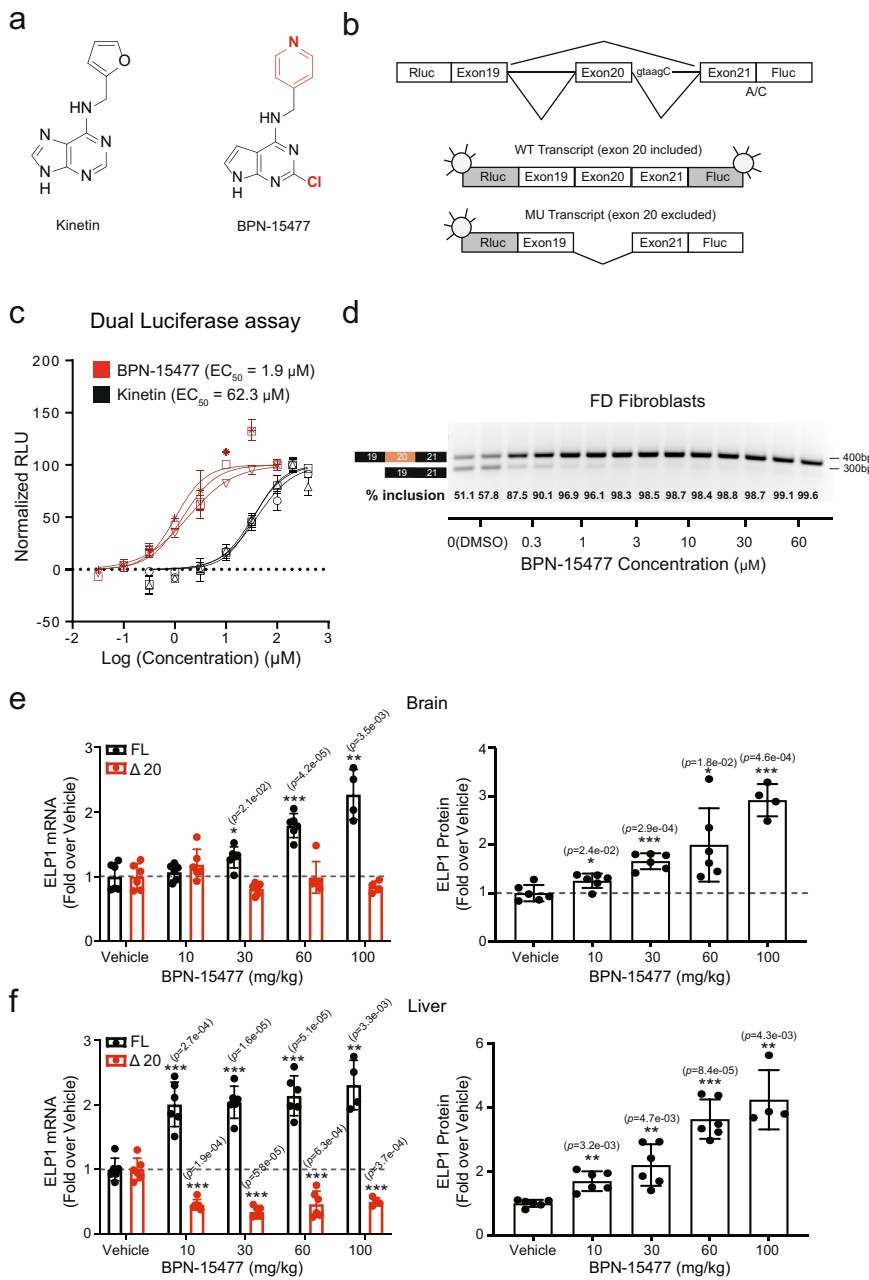

**Fig. 1 Identification of the small molecule splicing modulator BPN-15477. a** Molecular structure of BPN-15477 compared to kinetin, the northern heterocycle and C-2 substitution are indicated in red. **b** Schematic representation of the dual-reporter minigene used to measure splicing. Rluc and Fluc indicate Renilla and Firefly luciferase, respectively. A/C indicates the start codon mutation in Fluc and taagC indicates the location of the FD mutation. **c** Dose–response curves for kinetin and BPN-15477. Rluc-FD-Fluc transfected HEK293T cells treated for 24 h. Normalized relative luciferase units (RLU), which refer to the ratio between firefly and renilla luciferase and provide a measure of exon 20 inclusion, are plotted as a function of compound concentration. Assays were run in triplicate and curves were created by nonlinear regression using Prism4 ($n = 3$) (GraphPad Software Inc.). Data are presented as mean values ± SD. **d** Representative validation of BPN-15477 splicing correction in FD fibroblasts. Cells were treated for 24 h at the concentrations indicated. We independently repeated this experiment six times ($n = 6$). **e**, **f** Relative expression of full-length (FL) and Δ20 *ELP1* mRNA (left panel), and ELP1 protein quantification (right panel) in brain and liver after oral doses of BPN-15477 ranging from 10 to 100 mg/kg in adult transgenic *TgFD9* mouse ($n = 4$ mice in the 100 mg/kg treatment group and $n = 6$ mice in the other groups). Comparisons are done within the same color-coded group, against the vehicle-treated mice under two-tailed Welch's *t*-test. Data are presented as mean values ± SD. The unadjusted *p* values are displayed. In the figure, *$p < 0.05$; **$p < 0.01$; ***$p < 0.001$.

inclusion-response set (254 exon triplets), exclusion-response set (680 exon triplets), and the unchanged-response set (382 exon triplets with two expressed isoforms, $\Delta\psi < 0.01$ and FDR ≥ 0.1, Supplementary Data 1). We randomly divided these sequences into three non-overlapping groups, namely the training set (70%), the validation set (20%), and the test set (10%, see "Methods").

The network consisted of two layers of convolutions with a total of 2.5 million trainable parameters (Supplementary Fig. 2a) and was optimized for predicting splicing changes in every given exon triplet after BPN-15477 treatment. Our model achieved an average area-under-the curve (AUC) of 0.85 (Supplementary Figs. 2b–c). To evaluate the reproducibility of our model, we first

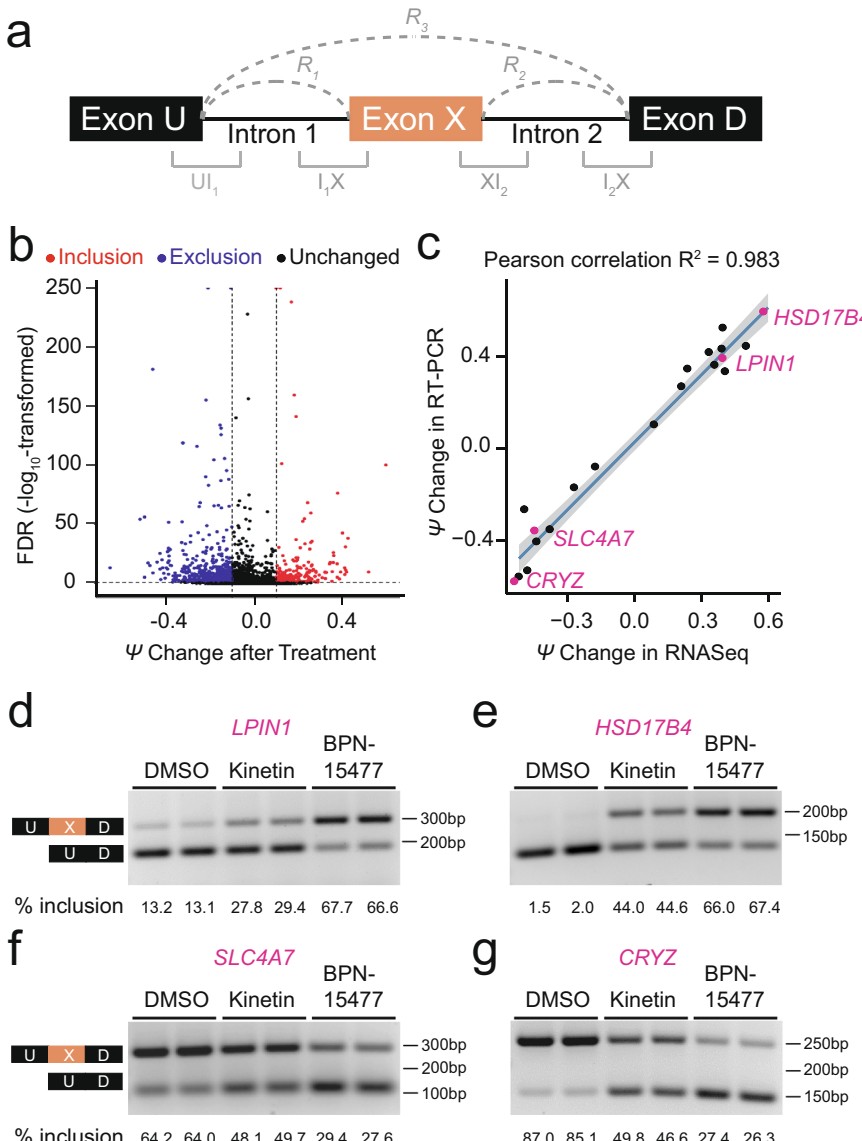

**Fig. 2 Transcriptome-wide changes in response to BPN-15477. a** Schematic representation of an exon triplet. $R_1$, $R_2$, and $R_3$ represent RNASeq reads spanning two adjacent exons. The region marked $UI_1$, $I_1X$, $XI_2$, and $I_2D$ represent 25 exonic base pairs and 75 intronic base pairs. **b** Volcano plot showing the $\psi$ changes after treatment with BPN-15477. Each dot represents one of the 161,097 expressed exon triplets in human fibroblasts. The x-axis represents the $\psi$ change after the treatment and the y-axis represents the false discovery rate (FDR) ($-$log10 transformed). Red dots represent the exon-triplets with an increase of middle exon inclusion ($\Delta\psi \geq 0.1$ and FDR < 0.1) while blue dots represent an increase of middle exon exclusion ($\Delta\psi \leq -0.1$ and FDR < 0.1). Black dots are exon-triplets not responsive to the treatment. The two vertical dashed lines indicate a $\psi$ change of 0.1 and $-0.1$ as thresholds for exclusion and inclusion, respectively. The horizontal dashed line indicates an FDR of 0.1. **c** Independent RT-PCR validation of splicing changes of twenty randomly selected candidates. For each validated exon-triplet, $\psi$ change measured by RNASeq (x-axis) is plotted against the splicing changes measured by RT-PCR (y-axis). The $R^2$ value indicates the coefficient of Pearson correlation. The solid line shows the estimated linear regression of the average changes. The gray zone indicates the 95% confidence interval for predictions from the estimated linear regression. For each exon-triplet we performed six independent replicates ($n = 6$). **d–g** Representative RT-PCR validations of splicing response of *LPIN1* (**d**), *HSD17B4* (**e**), *SLC4A7* (**f**), and *CRYZ* (**g**) in treated human fibroblasts. The upper bands indicate the isoform in which the middle exon is included while the lower bands indicate the isoform in which the middle exon is skipped. As expected, BPN-15477 is more potent than kinetin. For each gene in (**d–g**), we independently repeated the experiment six times ($n = 6$).

implemented random initialization of our training process 1000 times and found that the performance of all models was tightly distributed and aligned with our original model (Supplementary Fig. 2d). In addition, we found all of the top 10 first-layer filters contributing to the performance of our CNN model were highly correlated with those of the 1000 random-initialized models (average Pearson correlation $R^2 = 0.55$, Supplementary Fig. 2e), suggesting that our deep learning framework was robust. We identified 39 5-mer motifs from the first layer of the CNN model that best explain drug responsiveness (Supplementary Fig. 3).

Note, the treatment response was not determined by any of these motifs independently. Instead, the CNN model utilized the synergistic effect of all the motifs on a given sequence to make the classification decision. Thirteen of these motifs explained 92.62% of the AUC, each of which altered more than 0.05 of AUC for at least one class of prediction (see "Methods", Fig. 3a). Importantly, analysis of these motifs revealed that the sequence in proximity to, and in many cases encompassing, the 5′ splice site of the middle exon had the largest influence in modulating treatment response (see "Methods", Fig. 3b). These results emphasized the

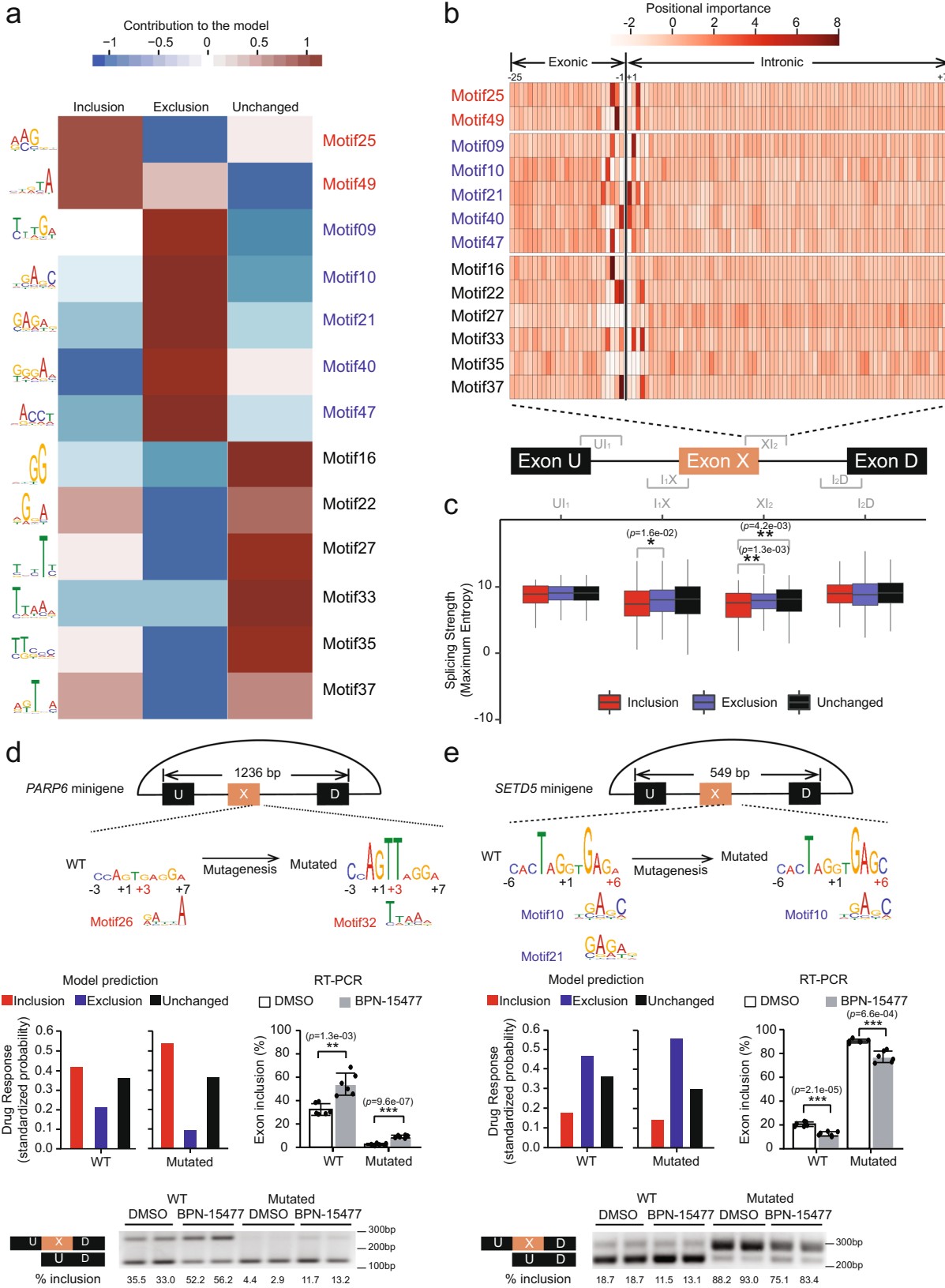

importance of the 5′ splice site in determining treatment outcome. In silico saturation mutagenesis (see "Methods") further supported these findings, revealing that base contribution to the treatment outcomes peaked around the 5′ splice sites of the middle exons, with distinct patterns amongst sequences with inclusion, exclusion and unchanged responses (Supplementary Fig. 4). To assess the robustness of the motifs identified by our CNN model, we compared the similarities of the 13 CNN-identified motifs with the most enriched five-nucleotide combinations (5-mer) at 5′ splice sites of the middle exons in our training set. The 5-mer sequence enrichment analysis for nucleotides at positions −3 to +7 of 5′ splice sites was highly

**Fig. 3 Convolutional neural network achieves high performance in predicting drug response and reveals sequence motifs that contribute to drug sensitivity. a** Heatmap of the top 13 motifs predicted to contribute to inclusion (red), exclusion (blue), or unchanged (black) response. The color bar indicates the directional contribution of each motif. The brown domain indicates positive contribution while the blue domain indicates negative contribution. The LOGO plot of each motif is shown on the left side of the heatmap. **b** Heatmap of motif importance at each position within $XI_2$ region at the 5′ splice site of the middle exon. The thick vertical line shows the exon–intron boundary. The bar indicates the positional importance, measured by positional activation in the first layer of the CNN model. **c** Box plot showing the strength of each splice junction along the exon triplets for inclusion (red), exclusion (blue), and unchanged (black) group defined by the RNASeq. The middle lines inside boxes indicate the medians. The lower and upper hinges correspond to the first and third quartiles. Each box extends to 1.5 times inter-quartile range (IQR) from upper and lower hinges, respectively. Outliers are not shown. The unadjusted $p$ values from two-tailed Welch's $t$-test are displayed. **d**, **e** Upper row: Schematic representations of *PARP6* and *SETD5* triplet minigene. The length of the exon triplets cloned into the minigenes is shown. The sequences adjacent to the 5′ splice site of the middle exon are shown in LOGO plots. The height of each nucleotide was estimated using in silico saturation mutagenesis (see "Methods"). The red coordinate numbers indicate the positions of mutations relative to the 5′ splice sites. Their closely matched CNN motifs are indicated beneath. Middle row: Splicing changes of the middle exons in both wild type and mutated exon triplets, predicted by the CNN model (left) and measured by RT-PCR of the minigene (right). To generate the bar plots of RT-PCR each experiment was repeated six times ($n = 6$, two-tailed Welch's $t$-test). Data are presented as mean values ± S.D. The unadjusted $p$ values are displayed. Bottom row: Example of splicing changes induced by the treatment using a minigene splicing assay. The percentage of middle exon inclusion is indicated beneath each lane. In the figure, $*p < 0.05$; $**p < 0.01$; $***p < 0.001$.

correlated with motifs identified by the CNN model (Supplementary Fig. 5a). We and others previously reported that SMCs with chemical structures similar to BPN-15477 promote, either directly or indirectly, the recruitment of U1 snRNP, to non-canonical 5′ splice sites, which have weak splice site strength scores[45,46]. Therefore, we evaluated splice site strength using MaxEntScan[55] and found that the 5′ splice site of the middle exon is significantly weaker when exon inclusion is enhanced by treatment (Fig. 3c). Intriguingly, the splice site strength did not distinguish the exon triplets that show an exclusion response.

We next sought to determine if our CNN model could predict treatment response in mutated exon triplets. We generated minigenes for treatment responsive alternatively spliced triplets in *PARP6*, *SETD5*, and *CPSF7*. These genes were chosen because their genomic triplet length enabled evaluation in an appropriate splicing vector. We mutated nucleotides in silico between positions $+2$ and $+6$ in the 5′ splice site of each triplet and used our CNN model to predict treatment response. Mutations predicted to be responsive were introduced into the minigenes and splicing was evaluated by RT-PCR. In all three minigene splicing assays, the RT-PCR data confirmed our CNN model predictions for mutant triplets (Fig. 3d, e, Supplementary Fig. 5b). Therefore, our CNN model will allow us to evaluate the potential therapeutic value of BPN-15477 on human splicing mutations.

**Identification of potential therapeutic targets of BPN-15477.** To evaluate the predictive power of our CNN model to determine which human disease-causing mutations might respond to treatment with BPN-15477, we first identified the pathogenic mutations that alter splicing in ClinVar. We considered all 89,642 annotated pathogenic or likely-pathogenic mutations (CV-pMUTs) and predicted their influence on splicing using SpliceAI[23]. We found that ~20% of all CV-pMUTs are predicted to alter splicing within 50 bp of the mutation, and that ~80% of these disrupt Ensembl-annotated (GRCh37 version 75) splice sites (Fig. 4a). We next used our CNN model to predict which genes harboring CV-pMUTs might respond to BPN-15477 treatment. We found the splice sites impacted by 14,272 CV-pMUTs from SpliceAI prediction exactly mapped to the splice sites of 11,616 exon triplets. Our CNN model predicted that 271 of these triplets should be responsive to BPN-15477 treatment (Supplementary Data 2) and identified 155 genes that harbor 214 annotated disease-causing mutations that could be targets for splicing correction using BPN-15477. The responsive genes containing the top 20 most frequent mutations (gnomAD v2.1.1) and their associated human diseases are shown in Table 1.

Examination of Table 1 demonstrates the remarkable therapeutic potential of this class of splicing modulators.

**Experimental validation of therapeutic targets for BPN-15477.** To evaluate whether BPN-15477 ameliorates aberrant splicing events, we searched for available human cell lines carrying the splicing mutations predicted to respond to our treatment (Supplementary Data 2). From the Coriell cell repository we were able to obtain a cell line with the c.894G > A mutation in the *LIPA* gene (Table 1)[56]. Mutations in *LIPA* cause both the severe infantile-onset Wolman disease and the milder late-onset cholesterol ester storage disease (CESD)[57–59] (MIM# 278000)[60]. The c.894G > A mutation leads to skipping of exon 8 and is responsible for the milder CESD. This is the most common *LIPA* gene mutation and it is found in 50% of individuals with lysosomal acid lipase (LAL) deficiency[61]. To validate the treatment effect on the splicing of exon 8, patient cells were treated with 60 µM of BPN-15477 for 24 h. As predicted, the treatment promoted the inclusion of exon 8, with mutated cells showing a 10% increase in normal transcript levels (Fig. 4b). To overcome the limitations in available patient cell lines harboring specific splicing mutations, we prioritized our experimental validations based on the availability of minigene constructs. Previous work has established that an expression minigene (EMG) system containing the full-length *CFTR* coding sequence is a reliable model for the study of variants that impact mRNA splicing[62–66]. To evaluate the efficacy of BPN-15477 to correct *CFTR* aberrant splicing caused by the c.2988G > A mutation (Supplementary Data 2), we generated a Flp-In-293 stable cell line expressing c.2988G > A CFTR-EMG-i14-i18 that contains full-length introns 14 and 16, and abridged introns 15, 17, and 18. The c.2988G > A variant is located in the last nucleotide position of exon 18 and results in a synonymous change (Gln996Gln) at the protein level. However, this variant also alters the 5′ splice donor site of intron 18 leading to exon 18 skipping[64]. RT-PCR using CFTR-specific primers revealed about 3% normal spliced transcript in the Flp-In-293 stable cells expressing 2988G > A (Fig. 4c). Treatment with BPN-15477 at 60 µM for 5 days increased exon 18 inclusion by 10% (Fig. 4c) confirming our CNN model prediction. We also generated a *MLH1* minigene spanning exons 16–18 harboring the c.1989G > A mutation (Supplementary Data 2) which leads to skipping of exon 17 and causes hereditary nonpolyposis colorectal cancer (HNPCC) or Lynch syndrome (MIM# 120435)[14,67]. As predicted, the treatment significantly increased exon 17 inclusion in cell lines expressing the mutated minigene (Fig. 4d, Supplementary Data 2).

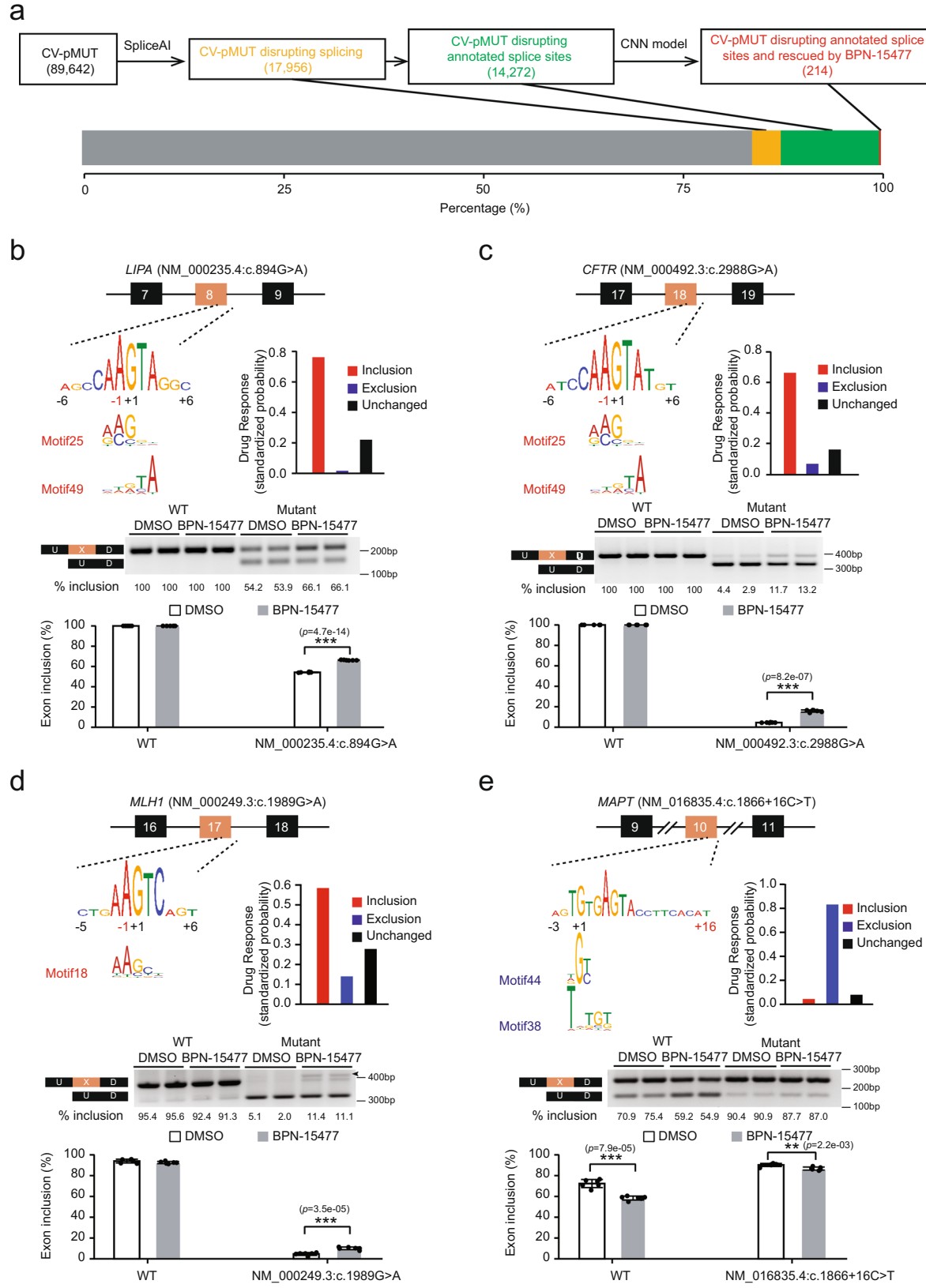

Finally, to validate the utility of our treatment to promote exon skipping as a therapeutic modality, we searched for drug responsive disease-causing mutations that lead to abnormal middle exon inclusion. The MAPT gene is associated with familial frontotemporal dementia and parkinsonism linked to chromosome 17 (FTDP-17, MIM# 600274)[68–71]. In healthy human brains, the alternative splicing of MAPT exon 10 is strictly regulated to maintain equal amounts of the 3R (exon 10 skipped) and 4R (exon 10 retained) tau isoforms. Disruption of this balance by increased 4R tau increases the risk of developing FTD. The c.1866 + 3G > A mutation, commonly referred to as IVS10 + 3, in the MAPT gene increases the inclusion of exon 10 and is

**Fig. 4 Identification of therapeutic targets for BPN-15477. a** Workflow to identify all potential therapeutic targets for BPN-15477. SpliceAI was applied on all ClinVar pathogenic mutations (CV-pMUTs) and the CNN model was used to determine whether CV-pMUTs disrupting annotated splice sites would be rescued by BPN-15477 treatment. The bar plot shows the percentage of each CV-pMUT category. **b–e** Upper row: the sequences at the 5′ splice site of *LIPA* Exon 8 in patient cells (**b**) and minigene constructs for *CFTR*, *MLH1*, and *MAPT* (**c–e**) are shown. The sequences around the 5′ splice sites of the middle exon are shown in LOGO plots, with closely matched CNN motifs indicated below. The red coordinate numbers indicate the position of mutations introduced mutations. The bar plots demonstrate the CNN model prediction of treatment response for the mutated sequences. Middle row: RT-PCR validation of treatment responses in cell lines carrying the specific splice mutation or the mutated minigene. The band marked by arrowhead in (**d**) was verified by sequence analysis that showed usage of a cryptic 5′ splice site at +31. To generate the bar plots, each experiment was repeated six times. Bottom row: The bar plots demonstrate the splicing change promoted by BPN-15477 treatment ($n = 6$). Data are presented as mean values ± S.D. The statistical significance is determined via two-tailed Welch's $t$-test. The unadjusted $p$ values are displayed. In the figure, $*p < 0.05$; $**p < 0.01$; $***p < 0.001$.

**Table 1 ClinVar pathogenic mutations predicted to be rescued by BPN-15477 treatment and selected based on top populational allele frequencies in gnomAD (v2.1.1).**

| Gene | Mutation | Frequency | Molecular consequence (ClinVar) | Splicing (SpliceAI) | Predicted drug response | Disease |
|------|----------|-----------|-------------------------------|---------------------|------------------------|---------|
| IL36RN | c.115 + 6T > C | 1.00E−03 | Intronic | Loss | Inclusion | Pustular psoriasis |
| LIPA | c.894G > A | 8.28E−04 | Synonymous | Loss | Inclusion | Lysosomal acid lipase deficiency |
| DNAH9 | c.1970 + 4A > G | 3.18E−04 | Intronic | Loss | Inclusion | Ciliary dyskinesia |
| CA5A | c.555G > A | 1.67E−04 | Synonymous | Loss | Inclusion | Carbonic anhydrase VA deficiency |
| ORC6 | c.449 + 5G > A | 1.63E−04 | Intronic | Loss | Inclusion | Meier-Gorlin syndrome 3 |
| SRD5A2 | c.547G > A | 1.59E−04 | Missense | Loss | Inclusion | 3-Oxo-5 alpha-steroid delta 4-dehydrogenase deficiency |
| DGUOK | c.591G > A | 1.27E−04 | Intronic/synonymous | Loss | Inclusion | Mitochondrial DNA-depletion syndrome 3 |
| HBB | c.92G > A | 1.03E−04 | Missense | Loss | Inclusion | Beta thalassemia |
| PIGN | c.963G > A | 8.31E−05 | Synonymous | Loss | Inclusion | Multiple congenital anomalies-hypotonia-seizures syndrome 1 |
| NPHP1 | c.1027G > A | 6.37E−05 | Missense | Loss | Inclusion | Nephronophthisis |
| KDSR | c.879G > A | 6.37E−05 | Synonymous | Loss | Inclusion | Erythrokeratodermia variabilis et progressive 4 |
| SLC12A1 | c.1942G > A | 6.37E−05 | Missense | Loss | Inclusion | Bartter syndrome, type 1 |
| NSD1 | c.6152-5T > G | 5.96E−05 | Intronic | Loss | Inclusion | Beckwith-Wiedemann syndrome |
| PARN | c.659 + 4_659 + 7delAGTA | 4.63E−05 | Intronic | Loss | Inclusion | Dyskeratosis congenita |
| GLA | c.639 + 919 G > A | 4.53E−05 | Intronic | Gain | Exclusion | Fabry disease |
| ATM | c.2250G > A | 4.39E−05 | Synonymous | Loss | Inclusion | Ataxia-telangiectasia syndrome |
| POLG | c.3104 + 3A > T | 3.98E−05 | Intronic | Loss | Inclusion | Progressive sclerosing poliodystrophy |
| GPX4 | c.476 + 5G > A | 3.65E−05 | Intronic | Loss | Inclusion | Spondylometaphyseal dysplasia |
| GYPA | c.232G > A | 3.60E−05 | Missense | Loss | Inclusion | Blood group erik |
| MYO7A | c.2904G > A | 3.20E−05 | Synonymous | Loss | Inclusion | Usher syndrome, type 1 |

associated with FTDP-17. Our CNN model predicted that the mutated sequence would be responsive to BPN-15477 and the treatment would promote exon 10 exclusion, thereby potentially restoring 3R/4R balance (Supplementary Data 2). We generated HEK293 cells stably expressing a *MAPT* minigene encompassing exons 9–11 and partial introns 9 and 10 with both the wild type sequence and the c.1866 + 3G > A mutation. As predicted, treatment with BPN-15477 led to a significant reduction of exon 10 inclusion in the wild-type MAPT transcript. Unfortunately, introduction of the c.1866 + 3G > A mutation in the minigene context completely disrupted exon 10 skipping in both the DMSO and BPN-15477 treated minigenes (Supplementary Fig. 5c). However, it is known that in the human brain the c.1866 + 3G > A mutation "leaks" a detectable amount of 3R tau isoform[72], suggesting that treatment with a splicing modulator in vivo might still increase the level of 3R tau.

Given the importance of identifying therapeutics that modulate the 3R/4R tau ratio, combined with our demonstration that BPN-15477 can promote skipping of MAPT exon 10, we next evaluated the most common MAPT mutation, a C to T substitution in intron 10, c.1866 + 16C > T, commonly called IVS10 + 16. Although this mutation increases inclusion of exon 10[68], it did not appear in our list of responsive mutations because SpliceAI

did not correctly predict its effect on splicing. We applied our CNN model to the MAPT c.1866 + 16C > T exon triplet, which predicted that BPN-15477 would promote exon 10 exclusion in the mutated triplet. Generation and evaluation of both wild-type and mutant minigenes confirmed our prediction (Fig. 4e).

Taken together, these results suggest that BPN-15477 and other SMCs of this class might be an effective therapy for reducing the inclusion of exon 10 in MAPT, thereby delaying or ameliorating disease symptoms in patients carrying mutations that disrupt the balance of exon 10 inclusion. Our success in validating several of our CNN model predictions in both human cells and minigenes demonstrates that deep learning methods can be effectively applied to identify therapeutic targets for known splicing modulators, and further sets the stage for evaluating their effectiveness across a wide range of human genetic diseases.

**BPN-15477 treatment increases LIPA and CFTR protein in disease-relevant cellular models.** Previously, we have shown that splicing correction of *ELP1* results in an increase of functional protein that can rescue neurologic phenotypes in an FD mouse model[50]. Further, other splice-correcting therapies for SMA, including ASOs (nusinersen) and small molecules (risdiplam),

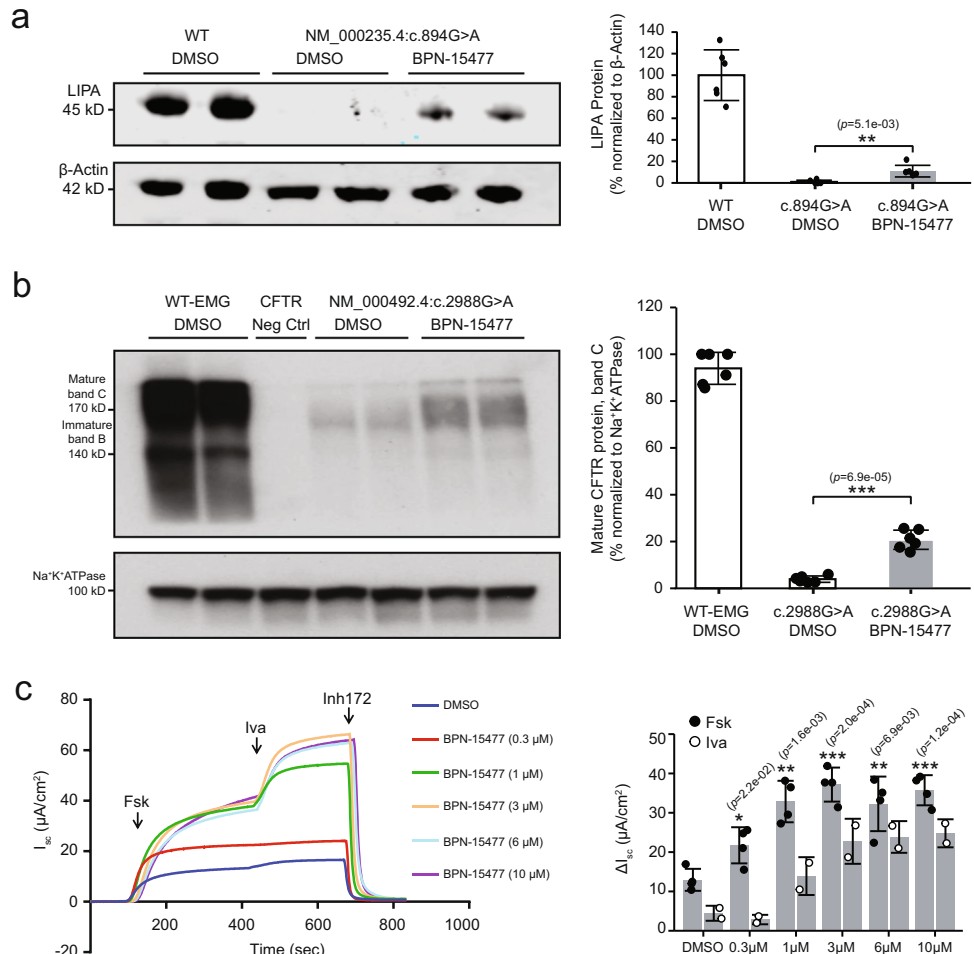

**Fig. 5 BPN-15477 treatment increases LIPA and CFTR protein in disease-relevant cellular models. a** Western blot analysis of LIPA protein in patient fibroblasts carrying the c.894G > A mutation. Left: Top and bottom panels show the blot probed with anti-LIPA and anti-ß-Actin antibody, respectively. Right: Bar chart showing the densitometric analysis of the western blot expressed as percentage of WT. LIPA was normalized to ß-Actin. To generate the bar plot, the experiment was independently repeated six times (n = 6). **b** Western blot analysis of CFTR protein in 293-Flpin cells stably expressing WT-EMG-i14-i18 or c.2988G > A-EMG-i14-i18. 293-Flpin cells with no endogenous expression of CFTR protein served as negative control. Left: Top and bottom panels show the blot probed with anti-CFTR and anti-Na$^+$K$^+$ATPase antibody, respectively. Right: Bar plot showing the densitometric analysis of the western blot expressed as percentage of mature CFTR protein, band C. Amount of mature CFTR protein was normalized to Na$^+$K$^+$ATPase. To generate the bar plot, the experiment was independently repeated six times (n = 6). In (**a**) and (**b**), data are presented as mean values ±S.D. The statistical significance is determined via two-tailed Welch's *t*-test. The unadjusted *p* values are displayed. **c** CFTR chloride channel analysis in CFBE-Flpin cells stably expressing c.2988G > A-EMG-i14-i18. Left: A representative tracing of short-circuit current ($I_{sc}$) measurements recorded in Ussing chambers after treatment of cells with either DMSO (vehicle) or variable doses of BPN-15477 for 72 h, as indicated on the figure labels. Cells were mounted on Ussing chambers to measure CFTR mediated chloride channel. After stabilization of the basal current, forskolin (10 µM) was added to the basolateral chambers followed by CFTR potentiator, Ivacaftor (10 µM), and CFTR Inhibitor 172 (10 µM) added to the apical chambers. Right: the bar plot indicates recovery of CFTR function upon treatment of cells with BPN-15477. Change in $I_{sc}$ ($\Delta I_{sc}$), a measure of CFTR function, was defined as the current inhibited by Inh-172 after sustained $I_{sc}$ responses were achieved upon stimulation with forskolin (Fsk, n = 4 $I_{sc}$ measurements per treatment) alone or sequentially with ivacaftor (Iva, n = 2 $I_{sc}$ measurements per treatment). Data are presented as mean values ± SD. The statistical significance is determined via two-tailed Welch's *t*-test when forskolin-stimulated CFTR function was compared between BPN-15477 treated and DMSO (vehicle) treated cells. The unadjusted *p* values are displayed. In the figure, *$p < 0.05$; **$p < 0.01$; ***$p < 0.001$.

show a direct correlation between splice correction and an increase in functional protein. To demonstrate that the same is true for a subset of our predicted targets, we treated patient fibroblasts carrying the major *LIPA* splicing mutation, c.894G > A, with 60 µM of BPN-15477 for 7 days. The treatment led to a 10% increase in functional LAL enzyme in mutated cells (Fig. 5a). Considering that a 3% increase in residual LAL enzyme activity is enough to distinguish Wolman disease, which is lethal in infancy, from the much milder CESD[60], a 10% increase in functional LAL would be predicted to have clinical benefit. Similarly, several published studies have shown that a small increase in CFTR

function translates into improved lung function and survival for individuals with cystic fibrosis (CF, MIM# 219700), even in moderate to advanced stages of disease[73–78]. CFTR-EMG expressing stable cell lines are an effective in vivo experimental system for evaluating the effectiveness of splicing modulation on CFTR protein production and chloride channel function[66]. To this end, we first assessed CFTR protein levels in Flp-In-293 cells stably expressing the c.2988G > A splicing variant after treatment with BPN-15477 for 5 days. Examination of Fig. 5b shows that the WT CFTR EMG_i14-i18 control cell line produces predominantly the higher molecular weight, complex-glycosylated

mature protein (band C ~ 170 kDa) as well as some lower molecular weight, core-glycosylated immature protein (band B). Flp-In-293 c.2988G > A stable cells treated with DMSO produce ~3% of WT complex-glycosylated mature CFTR protein while treatment with 60 μM BPN-15477 for 5 days increases the amount of mature CFTR protein to ~20% of WT (Fig. 5b).

Last, we tested the ability of BPN-15477 to rescue chloride channel function. We created CF bronchial epithelial (CFBE) cell lines that stably express the splicing mutation c.2988G > A. Cells were grown in monolayers on filters and treated with increasing doses of BPN-15477 (0.3–10 μM) or DMSO for 3 days. Chloride channel function was assessed by measuring short-circuit current ($I_{sc}$) on treated CFBE cells. Forskolin was added to initiate CFTR channel activity via cAMP-mediated signaling, with further channel activation by Ivacaftor and finally inhibition with Inh-172, a CFTR-specific inhibitor (Fig. 5c, representative $I_{sc}$ tracing). CFTR-specific change in current ($\Delta I_{sc} \pm SD$) allows for measurement of chloride channel function (Fig. 5c). Residual CFTR channel activity was observed in DMSO treated CFBE stable cells expressing c.2988G > A ($\Delta I_{sc} = 12.3 \pm 2.7$ μA/cm$^2$, Fig. 5c), which is consistent with previous reports showing that the c.2988G > A variant is associated with abnormal CFTR function and causes a mild form of CF[79]. Significant recovery of CFTR function (~3-fold) was observed following treatment with BPN-15477 at 1, 3, 6, and 10 μM for 3 days, with a maximal increase in CFTR function achieved using 3 μM of BPN-15477 ($\Delta I_{sc} = 37.168 \pm 4.32$ μA/cm$^2$, Fig. 5c). Importantly, the acute addition of Ivacaftor resulted in ~2-fold improvement in CFTR function in BPN-15477 treated cells. These results show that BPN-15477 treatment alone increased chloride channel function to ~20% of WT, and to ~30% of WT in combination with Ivacaftor[66]. As expected, the improvement in chloride channel function mediated by BPN-15477 correlates with a significant increase in exon 18 inclusion (Supplementary Fig. 6). Given that slight residual CFTR function is known to lead to mild CF, the increase in chloride channel function by treatment with BPN-15477 would be predicted to be clinically significant.

## Discussion

The development of drugs that can increase the amount of normal transcript through modulating RNA splicing in patients is a precisely targeted treatment approach aimed directly at the primary molecular disease mechanism without altering the genome. The recent success of splicing modulation therapies for DMD (exondys 51) and SMA (nusinersen, risdiplam rg7800, branaplam)[42,43] has validated the utility of splicing modification as a valuable therapeutic strategy for human disorders. Here, we have identified a compound, BPN-15477, that corrects splicing of *ELP1* in a minigene system and in FD patient cell lines. Further, in vivo splicing correction of *ELP1* in a humanized transgenic mouse model leads to an increase of ELP1 protein in all tissues, including brain, at a level that would be therapeutic since we have previously shown that even a small increase in functional protein has a dramatic effect on neurologic disease phenotype[50].

To determine the potential of the tool compound BPN-15477 to correct splicing of other genes, we developed a machine learning approach that uses sequence signatures to predict targetable splicing defects. Our CNN model identified a total of 39 5-mer motifs important for drug response, with 13 motifs accounting for most of the BPN-15477 sensitivity when motifs are located close to the 5′ splice site. Note, our model achieved an acceptable performance level (Supplementary Fig. 2b) with a training set of ~1000 non-redundant events, which is a relatively small size for a typical deep learning problem. This might be due to the following two reasons. First, we observed that the trained

model scored highly with a short region of ~8 bp (out of a 400 bp training region) flanking the 5′ ss of the middle exon, while the scores of other regions were almost zero (Supplementary Fig. 4). This suggests the effect site of the treatment might be in a restricted region relative to 5′ ss, and we might be able to train a simplified model on shorter sequences with fewer parameters. Second, we empirically avoided overfitting by applying the L1-regularization (coefficient = 0.6) in the convolutional layers and the dropout strategy (see "Methods") in the hidden layer during model training. Both of these factors made the model converge quickly (in 12 epochs, Supplementary Fig. 2b) on a small-sample size.

Evaluation of splice site strength in drug-responsive triplets where middle exon inclusion is increased showed that these exons have weaker 5′ splice sites, a finding consistent with previous studies[45,46,80]. We have reported that the splicing defect characteristic of FD is due to the weak definition of *ELP1* exon 20 and the kinetin-analog RECTAS was shown to promote the recognition of *ELP1* exon 20 through recruitment of U1 snRNP at the 5′ splice site[45,80]. Our CNN model predictions, combined with these previously published observations, suggest that our SMCs might act by promoting, either directly or indirectly, the recognition of weakly defined exons.

It is worth noting that, although the effect of BPN-15477 on the WT transcriptome was highly selective with only 0.58% of the expressed exon triplets responding to treatment, the major advantage of using a predictive model is that it allowed us to identify human disease-causing gene targets. Application of our CNN model to all ClinVar pathogenic mutations that disrupt splicing identified 214 human disease-causing mutations in 155 unique genes as potential therapeutic targets of BPN-15477, proving that deep learning models are a powerful approach to explore therapeutic targets for drugs that modify RNA splicing.

As proof of principle, we validated the treatment effect on splicing for several disease-causing mutations using patient cell lines or minigenes, and demonstrate the potential therapeutic feasibility of targeting splicing in patients with cystic fibrosis (*CFTR*), cholesterol ester storage disease (*LIPA*), Lynch syndrome (*MLH1*), and familial frontotemporal dementia (*MAPT*). These findings could have a significant impact for patients carrying these mutations. For example, Wolman disease and CESD are both caused by mutations in *LIPA*[57–59]. Wolman is lethal in infancy, whereas CESD patients have some residual enzyme activity and therefore have a milder clinical course. Remarkably, patients with only 3% of the normal level of *LIPA* transcript have the much milder disease CESD, and we show that BPN-15477 increases exon 8 inclusion and leads to a 10% increase in functional protein in a patient cell line, suggesting high potential therapeutic efficacy[60]. Similarly, several studies have shown that a small increase in functional CFTR protein is associated with a significant improvement in patient phenotype[73–78]. Therefore, our demonstration that BPN-15477 can correct splicing and increase chloride channel function to 20% of WT clearly suggests that our splicing-targeted approach is potentially efficacious.

The ability of BPN-15477 to promote the exclusion of exon 10 in the *MAPT* gene is particularly exciting given its role in frontotemporal dementia. Many FTDP-linked *MAPT* mutations alter the splicing of tau exon 10, generally leading to an increase in exon 10 inclusion and increased expression of 4R tau isoforms, thereby disrupting the 3R (exon 10 skipped) and 4R (exon 10 retained) tau ratio[68–71]. Herein we show significant exon 10 exclusion after treatment in both the WT and in the c.1866 + 16C > T minigene, which is the most common mutation associated with FTD in humans. From a therapeutic perspective, restoring the 3R/4R tau ratio has the potential to reduce or even eliminate unbound 4R tau. Our results further suggest that an

optimized form of this SMC might be beneficial for other forms of FTD caused by gain of function mutations in exon 10, such as P301L, P301S, or the S305N, since treatment could reduce the level of mutated transcript.

To our knowledge, this work represents the first application of a machine learning approach to analyze the global activity of a splicing modifier compound and identify therapeutic targets. Our successful laboratory validation of several of our CNN model predictions establishes the promise of such approaches and may presage the future advances in precision medicine offered by deep learning techniques. Although we recognize that our compound needs additional medicinal chemistry optimization for any specific indication in order to increase the potency and reduce the potential for off-target effects, this study provides a unique way in which therapeutic targets for small molecule splicing modulators can be discovered and significantly broadens their potential value in treating human genetic disease.

## Methods

**Preparation of BPN-15477 (2-chloro-N-(4-pyridylmethyl)-7H-pyrrolo[2,3-d] pyrimidin-4-amine)**. BPN-15477 was manufactured by Albany Molecular Research Inc. (AMRI) and composition of matter is covered in International application No. PCT/US2016/013553. Synthesis and analytical data for BPN-15477 are described below. All materials used in the studies were >99% pure, as assessed by analytical methods including NMR, HPLC, and LC/MS.

To a stirred suspension of 2,4-dichloro-7H-pyrrolo[2,3-d]pyrimidine (**1**, 5001 mg, 26.60 mmol, 1.000 eq.), obtained from AstaTech Inc., Bristol, PA, in 1,4-dioxane (50.0 mL) was added 4-(aminomethyl)pyridine (**2**, 3420 mg, 3.20 mL, 31.6 mmol, 1.19 eq.) followed by N,N-diisopropylethylamine (4450 mg, 6.00 mL, 34.1 mmol, 1.28 eq.) at room temperature. The reaction mixture was then heated to 90 °C and stirred at that temperature overnight.

The reaction progress was monitored by LC-MS analysis of an aliquot of the reaction mixture. After 12 h, ~6% starting material was detected by LC-MS. The reaction was quenched by water resulting in an emulsion. The mixture was filtered through Celite and then washed well with EtOAc ($3 \times 80$ mL). The organic phase was separated and the aqueous phase was extracted with EtOAc ($3 \times 40$ mL). The combined organic phases were washed with brine (50 mL) and then dried over sodium sulfate.

The volatiles were removed under reduced pressure to give the crude product as dark brown solid. To the crude solid was added EtOAc (100 mL). The mixture was then heated at reflux for 15 min before it was slowly cooled to room temperature. The resulting precipitate was collected by filtration, then washed well with cold EtOAc (30 mL) followed by diethyl ether (100 mL). The solid was dried under high vacuum overnight to afford 2-chloro-N-(4-pyridylmethyl)-7H-pyrrolo[2,3-d] pyrimidin-4-amine (BPN-15477) as a light brown solid (3450 mg, 13.3 mmol, 50% yield).

LC-MS: 0.63 min (254 nm), $m/z$ 260.3, 262.3 $[M + H]^+$, 258.2, 260.2 $[M-H]^-$; $^1$H NMR (DMSO-$d_6$) $\delta$: 11.65–11.85 (m, 1H), 8.51 (d, J = 6.0 Hz, 2H), 8.45–8.50 (m, 1H), 7.28–7.40 (m, 2H), 7.09–7.21 (m, 1H), 6.53–6.74 (m, 1H), 4.61–4.81 (m, 2H).

The identity and purity of the compound were established through $^{13}$C-NMR (Supplementary Fig. 7).

$^{13}$C-NMR (500 MHz, METHANOL-$d_4$): $\delta$ 158.73, 154.58, 152.19, 152.05, 150.06, 124.18, 123.05, 102.99, 99.98, 44.23.

**Cell culture**. HEK-293T (ATCC) cells were cultured in Dulbecco's modified Eagle's medium (11995-065, D-MEM, Gibco) supplemented with 10% fetal bovine serum (FBS, 12306C, Sigma) and 1% penicillin/streptomycin (30-009-CI, Corning).

Flp-In 293 cells (R75007, ThermoFisher Scientific) stably transfected with the *CFTR* expression minigenes (EMGs)[64] or HEK-293T stably transfected with *MAPT* minigene were cultured in D-MEM supplemented with 10% FBS, 1% penicillin/streptomycin, and 0.1 mg/mL Hygromycin (400052-5ML, Sigma).

Patient human fibroblast GM04663 (Coriell Cell Repository) carrying the c.2204 + 6T > C mutation in *ELP1*, GM03111 (Coriell Cell Repository) carrying the c.894G > A mutation in *LIPA*, and the human wild-type fibroblasts (Coriell Cell Repository) listed in Supplementary Table 1 used for RNA sequence were cultured in D-MEM supplemented with 10% FBS and 1% penicillin/streptomycin.

**Treatment of cultured cells**. Different batches of BPN-15477 were used for dissolution into 100% DMSO to yield 40 mM stock solutions. Working solutions (10X) were prepared by dilution in 5% DMSO in phosphate-buffered saline (PBS, 10010023 ThermoFisher Scientific). The final DMSO concentration in the treated or untreated cells was 0.5%. Kinetin was purchased from Sigma (K3253).

Cells to be treated with BPN-15477 are seeded at the appropriate density in specific vessels so as to reach semiconfluency at the time of treatment. HEK293T

transfected with minigenes were seeded in 6 wells and patient fibroblasts in 10-cm dishes using the described media. The following day, the media was changed with regular growth media supplemented with compound or DMSO working solutions to obtain final concentrations of 60 μM BPN-15477 and 0.05% DMSO. 60 μM BPN-15477 was chosen to guarantee the maximum effect possible on splicing (Fig. 1b and c). Cells were collected for RNA extraction 24 h after compound or DMSO addition.

**Transfection**. HEK293T cells were seeded in 6-well culture plates at $1.20 \times 10^6$ cells/well in D-MEM, 10% FBS, without antibiotics and incubated overnight to reach ~90% confluence the day after. Transfection was performed with FuGENE® HD Transfection Reagent (E2311, Promega) using the FuGENE-DNA ratio at 3.5:1 and following manufacturer protocol. After 4 h of incubation at 37 °C, cells were plated at a density of $3 \times 10^4$ cells/well in a poly-L-lysine-coated 96-well plate for the dual-luciferase assay or at the density of $8.5 \times 10^5$ cells/well into 6-well plates for minigene transfection. After 16 h incubation at 37 °C, SMCs or DMSO were added at the desired concentrations as described above and kept in culture for other 24 h.

**Dual-luciferase splicing assay**. Rluc-FD-Fluc plasmid used for the dual-luciferase splicing assay was derived using the *ELP1* minigene[44] containing the *ELP1* genomic sequence spanning exon 19–21 inserted into spcDNA3.1/V5-His Topo (Invitrogen). Firefly luciferase (FLuc) coding sequence was inserted immediately after exon 21 and renilla luciferase (RLuc) upstream of exon 19. Characterization of the assay has shown that RLuc is expressed each time a transcript is generated from the reporter plasmid, while FLuc is only expressed if exon 20 is included in the transcript, thereby keeping FLuc in-frame. Evaluation of FLuc/RLuc expression yields the percent exon inclusion in the splicing assays[49]. To perform the dual-luciferase assay HEK-293T were transfected with the described plasmid and treated with SMCs for 24 h as illustrated above. After treatment the cells were washed once in PBS and lysed for 25 min at room temperature using 50 μL well of passive lysis buffer (E1941, Promega). Luciferase activity was measured in 20 μL of cell lysate using the Dual-Luciferase Reporter Assay reagents (E1960, Promega) and the GloMax 96 Microplate Luminometer[49]. The integration time on the luminometer was set at 10 s. BPN-15477 was serially diluted in DMSO and PBS to generate a concentration–response curve over eight concentrations. The final concentration of DMSO in the media was kept at 0.5%. Cells cultured in the presence of 0.5% DMSO were used as a control.

**RNA isolation and RT-PCR analysis**. After treatment, cells were collected and RNA was extracted with QIAzol Lysis Reagent (79306, Qiagen) following the manufacturer's instructions. The yields of the total RNA for each sample were determined using a Nanodrop ND-1000 spectrophotometer.

Reverse transcription was performed using 0.5–1 μg of total RNA, Random Primers (C1181, Promega), Oligo(dT)15 Primer (C1101, Promega), and Superscript III reverse transcriptase (18080093, ThermoFisher Scientific) according to the manufacturer's protocol. cDNA was used to perform PCR reaction in a 20–25 μL volume, using GoTaq® green master mix (MT123, Promega). Primers and melting temperature ($T_m$) used are described (see Supplementary Methods). To measure the splicing of the minigenes, forward and reverse primers were designed to include the TOPO/V5 plasmid vector and flanking exon sequence in order to avoid endogenous gene detection. PCR reaction was performed as follows: 32 cycles of (95 °C for 30 s, $T_m$ for 30 s, 72 °C for 30 s), products were resolved on a 1.5–3% agarose gel, depending on the dimension of the bands to be separated, and visualized by ethidium bromide staining.

Ratios between isoforms with included or excluded middle exon were obtained using the integrated density value (IDV) for each correspondent band, assessed using Alpha 2000TM Image Analyzer and quantified by ImageJ software. The level of exon inclusion was calculated as the relative density value of the band representing inclusion and expressed as a percentage.

**Protein isolation and western blot analysis**. Protein extracts were obtained by homogenizing cells in RIPA buffer (Tris–HCl 50 mM, pH 7.4; NaCl 150 mM; NP-40 1%; sodiumdeoxycholate 0.5%; SDS 0.1%, 1 mM DTT) containing protease and phosphatase inhibitor cocktail (Roche). Insoluble debris were discarded after centrifugation and protein concentration was determined using Pierce® BCA Protein Assay Kit (Thermo Scientific). For LIPA WB, 30 μg of protein lysate was separated on NuPage 10% Bis–Tris gel (Invitrogen) and transferred on nitro-cellulose membrane (Thermo Scientific). Membrane was blocked in Odyssey blocking buffer (Licor Biosciences) for 1 h at room temperature and incubated overnight at 4 °C with the mouse monoclonal antibody against LIPA (Abnova clone 9G7F12, 1:200) and with the rabbit polyclonal antibody against beta actin (A2066, Sigma, 1:5000). Membranes were washed three times in PBS with 0.1% tween 20 and incubated with IRDye secondary antibodies (Licor Biosciences) for 1 h at room temperature. Protein bands were visualized by Odyssey CLx imager (Licor Biosciences).

For CFTR WB, 40 μg of protein lysate was separated on 7.5% Criterion TGX protein gel (BioRad). Transfer to PVDF membrane was performed in a Trans-Blot Turbo Transfer System (BioRad). After blocking in 5% non-fat dry milk (BioRad),

the membrane was probed with mouse monoclonal anti-CFTR antibody that recognizes amino acids 1204-1211 (596, CFFT, North Carolina Chapel Hill, 1:5000). Rabbit monoclonal anti-Na+K + ATPase (Abcam, 1:50,000) was used as a loading control. Secondary antibodies were anti-mouse (GE Healthcare, 1:150,000) and anti-rabbit (GE Healthcare, 1:100,000), respectively. Blots were exposed on film using ECL Primer Western Blotting Detection Reagent (GE Healthcare).

**CFTR functional assessment and response to BPN-15477.** Assessment of CFTR channel function and response to BPN-15477 was performed in Cystic Fibrosis Bronchial Epithelial cells (CFBEs) stably expressing c.2988G > A[66,81,82]. CFBEs were plated on snapwell filters cultured at 37 °C/5% $CO_2$ in Minimum Essential Medium (11095098 MEM, ThermoFisher Scientific) supplemented with 10% FBS (35-010-CV, Corning Cellgro), 1% Pencillin Streptomycin (15140122, Thermo-Fisher Scientific), and 100 µg/ml Hygromycin (10687010, ThermoFisher Scientific). When transepithelial resistance reached ~200 Ω (~5–7 days) as measured using Epithelial Volt/Ohm Meter (World Precision Instruments), cells were treated with BPN-15477 at varying doses (0.3–10 µM) for the next 3 days. Filters were mounted in Ussing chambers and $I_{sc}$ was measured with a VCC MC6 multichannel voltage-current clamp amplifier (Physiological Instruments). A high chloride solution (145 mM NaCl, 1.2 mM $MgCl_2$, 1.2 mM $CaCl_2$, 10 mM glucose, 10 mM HEPES, pH 7.4) was added to the basolateral chamber and a low chloride solution was added to the apical chamber. Buffers in both chambers were maintained at 37 °C and air was bubbled in to introduce circulation. After equilibration of currents, 10 µM forskolin (Selleckchem) was added to the basolateral side to activate CFTR channels via cAMP signaling. Currents were allowed to plateau, followed by acute addition of 10 µM ivacaftor at apical side for CFTR potentiation (Selleckchem), and finally, inhibition of CFTR-specific currents using 10 µM Inh-172 (Selleckchem) added to the apical chamber. A drop in short-current ($\Delta I_{sc}$), defined as the current inhibited by Inh-172 after sustained $I_{sc}$ responses were achieved upon stimulation with forskolin alone or sequentially with ivacaftor, was a quantifiable measurement assigned to CFTR channel function.

**BPN-15477 administration via oral gavage.** *TgFD9* transgenic mice were treated by oral gavage once daily for 7 days with BPN-15477 as a suspension in 0.5% HPMC, 0.1% Tween 80 at a dose of 10, 30, 60, or 100 mg/kg.

The mice used for this study were housed in the animal facility at Rutgers University, provided with access to food and water ad libitum, and maintained on a 12-h light/dark cycle at a temperature of 70 °F with 30–70% humidity, and all experimental protocols were approved by the Institutional Animal Care and Use Committee of the Rutgers University, and were in accordance with NIH guidelines. For routine genotyping, genomic DNA was prepared from tail biopsies and PCR was carried out to detect the *TgFD9* transgene using the following primers: forward, 5′-GCCATTGTACTGTTTGCGACT-3′; reverse, 5′-TGAGTGTCACGA TTCTTTCTGC-3′.

**RNA isolation and qRT-PCR analysis of full-length and mutant *ELP1* transcripts in mouse tissues.** Mice were euthanized and brain, liver, lung, kidney, heart, and skin tissues were removed and snap-frozen in liquid nitrogen. Tissues were homogenized in ice-cold QIAzol Lysis Reagent (Qiagen), using Qiagen TissueLyser II (Qiagen). Total RNA was extracted using the QIAzol reagent procedure provided by the manufacturer. The yield, purity, and quality of the total RNA for each sample were determined using a Nanodrop ND-1000 spectrophotometer. Full-length and mutant *ELP1* mRNA expression was quantified by quantitative real-time PCR (qRT-PCR) analysis using CFX384 Touch Real-Time PCR Detection System (BioRad). Reverse transcription and qPCR were carried out using One Step RT-qPCR (BioRad) according to the manufacturer's recommendations. The mRNA levels of full-length *ELP1*, mutant Δ20 *ELP1*, and *GAPDH* were quantified using Taqman-based RT-qPCR with a cDNA equivalent of 25 ng of starting RNA in a 20-µl reaction. To amplify the full-length *ELP1* isoform, FL *ELP1* primers forward, 5′-GAGCCCTGGTTTTAGCTCAG-3′; reverse, 5′-CATGCATTCAAATG CCTCTTT-3′ and FL *ELP1* probe 5′-TCGGAAGTGGTTGGACAAACTTATG TTT-3′ were used. To amplify the mutant (Δ20) *ELP1* spliced isoforms, Δ20 *ELP1* primers forward, 5′-CACAAAGCTTGTATTACAGACT-3′; reverse, 5′-GAAG GTTTCCACATTTCCAAG-3′ and Δ20 *ELP1* probe 5′-CTCAATCTGATTTAT GATCATAACCCTAAGGTG-3′ were used to amplify the mutant (Δ20) *ELP1* spliced isoforms. The *ELP1* forward and reverse primers were each used at a final concentration of 0.4 µM. The *ELP1* probes were used at a final concentration of 0.15 µM. Mouse *GAPDH* mRNA was amplified using 20X gene expression PCR assay (Life Technologies, Inc.). RT-qPCR was carried out at the following temperatures for indicated times: Step 1: 48 °C (15 min); Step 2: 95 °C (15 min); Step 3: 95 °C (15 s); Step 4: 60 °C (1 min); Steps 3 and 4 were repeated for 39 cycles. The Ct values for each mRNA were converted to mRNA abundance using actual PCR efficiencies. *ELP1* FL and Δ20 mRNAs were normalized to *GAPDH* and vehicle controls and plotted as fold change compared to vehicle treatment. Data were analyzed using the SDS software.

**Homogeneous time-resolved fluorescence (HTRF) assay for ELP1 protein quantification in mouse tissues.** Tissue samples were collected, snap-frozen in liquid nitrogen, weighed, and homogenized on the TissueLyzer II (Qiagen) in RIPA buffer (Tris-HCl 50 mM, pH 7.4; NaCl 150 mM; NP-40 1%; sodium deoxycholate 0.5%; SDS 0.1%) containing a cocktail of protease inhibitors (Roche) at a tissue weight to RIPA buffer volume of 50 mg/mL. The samples were then centrifuged for 20 min at $14,000 \times g$ in a microcentrifuge. The homogenates were transferred to a 96-well plate and were diluted in RIPA buffer to ~1 mg/mL for ELP1-HTRF and ~0.5 mg/mL for total protein measurement using the BCA protein assay (Pierce). Samples were run in duplicate and averaged. For the ELP1-HTRF assay, 35 µL of tissue homogenate were transferred to a 384-well plate containing 5 µL of the antibody solution (1:50 dilution of anti-ELP1 D2 and anti-ELP1 cryptate from Cisbio). The plate was incubated overnight at room temperature. Fluorescence was measured at 665 and 620 nm on an EnVision multilabel plate reader (Perkin Elmer). Total protein content was quantified in each tissue homogenate using the BCA assay according to the manufacturer's protocol. The total protein normalized change in ELP1 protein signal for BPN-15477 and vehicle-treated tissue sample was calculated as ratio of the signal in the presence of the test compound over the signal in the absence of the test compound (vehicle control).

**RNASeq experiment.** Six different human fibroblast cell lines from healthy individuals were obtained from Coriell Institute (Supplementary Table 1) and cultured in D-MEM supplemented with 10% FBS and 1% penicillin/streptomycin. Cells were counted and plated in order to achieve semi-confluence after 8 days. Twenty-four hours after plating, the medium was changed and cells were treated with BPN-15477 or DMSO to a final concentration of 30 µM and 0.5%, respectively. DMSO was used as vehicle and the concentration of BPN-15477 was chosen based on our previous studies, since at this concentration BPN-15477 induces robust splicing changes and ELP1 protein increase. The medium was changed after 3 days. These conditions have been previously established in our laboratory and the time points chosen to maximize differential protein expression. After seven days of treatment, cells were collected, and RNA was extracted using the QIAzol Reagent following the manufacturer's instructions. RNASeq libraries were prepared by the Genomic and Technology Core (GTC) at MGH using strand-specific dUTP method[83]. RNA sample quality (based on RNA Integrity Number, or RIN) and quantity were determined using the Agilent 2200 TapeStation and between 100 and 1000 ng of total RNA was used for library preparation. Each RNA sample was spiked with 1 µl of diluted (1:100) External RNA Controls Consortium (ERCC) RNA Spike-In Mix (4456740, ThermoFisher Scientific) alternating between mix 1 and mix 2 for each well in the batch. Samples were then enriched for mRNA using polyA capture, followed by stranded reverse transcription and chemical shearing to make appropriate stranded cDNA inserts. Libraries were finished by adding Y-adapters, with sample-specific barcodes, followed by between 10 and 15 rounds of PCR amplification. Libraries were evaluated for final concentration and size distribution by Agilent 2200 TapeStation and/or qPCR, using Library Quantification Kit (KK4854, Kapa Biosystems), and multiplexed by pooling equimolar amounts of each library prior to sequencing. Pooled libraries were 50 base pair paired-end sequenced on Illumina HiSeq 2500 across multiple lanes. Real-time image analysis and base calling were performed on the HiSeq 2500 instrument using HiSeq Sequencing Control Software (HCS) and FASTQ files demultiplexed using CASAVA software version 1.8. RNASeq reads were mapped to the human genome Ensembl GRCh37 by STAR v2.5.2a allowing 5% mismatch[84]. Exon triplet index was built according to transcriptome Ensembl GRCh37 version 75. Reads spliced at each exon triplet splice junction was calculated by STAR on the fly.

**Differential splicing analysis.** For each exon triplet in a certain biological replicate, $\psi$ was calculated according to Fig. 2a as $\psi = 0.5 * (R_1 + R_2)/(0.5 * (R_1 + R_2) + R_3)$.

The average $\psi$ was calculated for treated and untreated conditions, followed by the calculation of $\psi$ change. For a certain exon triplet in a certain biological replicate, a $2 \times 2$ table was created, where the four cells of the table represent number of reads supporting middle exon inclusion and skipping before and after treatment. Thus, for each exon triplet, totally six $2 \times 2$ tables were created for six biological replicates. Cochran–Mantel–Haenszel test was applied to test whether there is an association between treatment and splicing across all replicates (namely whether the cross-replicate odds ratio is 1 or not). For each exon triplet, a $p$ value of Cochran–Mantel–Haenszel test was reported. Benjamini–Hochberg false-discovery-rate (BH FDR) correction was finally applied to $p$ values of all triplets.

**CNN model.** Our CNN network contains two convolutional layers and one hidden layer (Supplementary Fig. 2a) and it was trained using Basset framework[41]. The training set consists of 178 inclusion-responded, 476 exclusion-responded, and 268 unchanged exon triplets. The validation sets consist of 51 inclusion-responded, 136 exclusion-responded and 76 unchanged exon triplets. The test set consists of 25 inclusion-responded, 68 exclusion-responded, and 38 unchanged exon triplets. The above three sets were assigned randomly in Python using seed of 122. For each exon-triplet, the sequences from $UI_1$, $I_1X$, $XI_2$, and $I_2D$, each of which consisted of 25 bp of exonic sequence and 75 bp of intronic sequence, are concatenated and then one-hot coded into an input matrix with size of $4 \times 400$. The first round of convolution is applied with fifty $4 \times 5$ weight matrices, converting the input matrix into a $50 \times 396$ convoluted matrix, in which each row represents the convolution of one weight matrix. Then the convoluted matrix is nonlinearly transformed by

rectified linear unit (ReLU) function and max pool stage takes the maximum of two adjacent positions of each row, shrinking the output matrix to a size of $50 \times 198$. Then the second round of convolution applies fifty $4 \times 2$ weight matrices, followed by the same ReLU transformation and max pool of the first round. The output is converted to $1 \times 500$ matrix to initiate the hidden layer, where a fully connected network is built with 90% dropout rate. The output from the hidden layer is ReLU transformed again and is then linearly transformed into a vector of three values, representing three different treatment responses. The final sigmoid nonlinearity maps each element in the vector to a value between 0 and 1, considering the probability of drug responsiveness. In each epoch of training, average of area-under-curve (AUC) was measured on the validation set across the prediction of three treatment responses. The training and validation loss in terms of binary cross-entropy were measured on the training set and validation set, respectively. The training process stopped if there is no improvement of the average of AUC in 10 consecutive epochs. In this study, we stopped at 12th epoch to avoid overfitting (Supplementary Fig. 2b).

**Evaluation model performance transcriptome-wide**. To examine the model performance on human transcriptome, all exon triplets whose PSI was between 0.01 and 0.99 either before or after treatment were recruited. The "true inclusion responding" group was defined triplets with PSI changes ≥0.01 after treatment while the "true exclusion responding" group was defined as triplets with PSI changes ≤−0.01 after treatment. The remaining ones were considered as "true unchanged" group. The CNN model was then applied on these triplets and ROC-AUC and PR-AUC were calculated respectively.

**Random initialization**. The initial parameters were generated from the random selection of 1000 different seeds. Each set of these initial parameter sets was used to train a CNN model of the same structure. The training, validation, and test rules were the same as our original model. By doing so, 1000 random CNN models were trained together with the original one in this study. For each model, AUC for each class was calculated.

**Examination of motif contribution**. To examine motif contribution in classification, the test set was used as model input. For each motif's contribution to be measured, its total output of the first convolutional layer was manually set to zero, mimicking the situation where that motif had not been found. The model was then taken forward without tuning other parameters and the new AUC of each class was calculated. The contribution of that motif was measured as the difference between the new AUC and the original AUC of each class. All motifs were investigated in this way in turn. A motif whose contribution is more than 0 in any class is considered a true identified motif. A motif whose contribution is no less than 0.05 in any class is considered a top contributor in drug response prediction.

**Analysis of positional importance**. To examine each motif contribution in classification, the test set was used as model input. We directly investigate the positional activation of the first layer of each motif. For each motif, we averaged the first layer activation of the same position across all sequences. Then we transformed the positional average activations of each motif to their z-scores.

**In silico saturation mutagenesis**. For a given input sequence, each position was mutated in silico to the other three alternative nucleotides. The loss of the model using mutated sequences was calculated and compared to the loss derived from the original sequence. The maximum change to the loss at each position was recorded. Nucleotides were drawn proportionally to the change of loss, beyond a minimum height of 0.25.

**Standardized probability from prediction**. To determine the final drug response (inclusion, exclusion, or unchanged) from the prediction, the raw prediction score from the model was standardized. For each class, a cutoff representing 95% specificity of that class was identified on the validation set. The intermediate score of each class was calculated as the raw prediction score divided by the cutoff of that class. The standardized probability for each class was then calculated as the intermediate score divided by the sum of intermediate scores of three classes.

***k*-mer enrichment analysis**. The sequences at −3 to +7 bp of the 5′ splice sites of the middle exons for inclusion, exclusion, and unchanged exon triplets were extracted. For each class, 5-mer enrichment was estimated against the other two classes using Discriminative Regular Expression Motif Elicitation (DREME) from THE MEME Suite with the parameter "*-p, -n, -dna -e 0.05 and -k 5*"[85]. The position probability matrix (PPM) of each motif derived from the 5-mer analysis was correlated to the PPM of each motif derived from the CNN model, using Pearson correlation.

**Splicing strength analysis**. Splice strength was measured by maximum entropy model[55]. As described in the original study, the measurement in this paper takes the short sequence of 9 and 23 bp flanking splice junctions, depending on whether it is 5′ or 3′ side.

**Candidate selection for minigene validation**. To validate whether the CNN model correctly predicts BPN-15477 treatment response of mutated exon triplets, the following rules were applied to select suitable exon triplet: (1) exon triplets with a total length, including introns, <1.5 kb suitable for cloning; (2) exon triplets whose splicing changes were detectable in fibroblast RNASeq after BPN-15477 treatment, so that their drug responses could be used as positive control for the mutated minigenes; (3) the minigene recapitulates the same splicing change measured in the fibroblast by RNASeq to guarantee that the splicing process is intact in the minigene.

**Minigene generation**. Wild-type and mutant double-stranded DNA (dsDNA) fragments, selected based on low nucleotide length and exon-skipping probability, were ordered through GENEWIZ (FragmentGENE). Adenosine was enzymatically attached to DNA fragment 3′ ends with Taq Polymerase in the presence of 200 nM dATP and 2 mM $MgCl_2$ at 70 °C for 30 min. Fragments were ligated into linearized pcDNA™3.1/V5-His TOPO® TA plasmid (K480001 ThermoFisher Scientific) according to manufacturer's instructions. After colony selection and sequence confirmation, each plasmid was finally purified using MIDIprep kit (740410, NucleoBond® Xtra Midi, Takara, Mountain View, CA). Concentrations were determined using nanodrop spectrometer.

**SpliceAI prediction on ClinVar pathogenic mutations**. The VCF file recording ClinVar (version 20190325) mutations was downloaded. The pathogenic/likely pathogenic mutations were extracted and fed to SpliceAI (https://github.com/illumina/SpliceAI). In the prediction from SpliceAI, any mutation with any SpliceAI score no less than 0.2 was considered alter splicing. Therefore, such mutation, together with its influenced splice junction (ISJ) and SpliceAI score were recorded.

**Rescue definition and prediction**. For an exon triplet, the coordinates of the two domains of the middle exon were compared to those ISJs discovered by SpliceAI. If either of the domain coordinates were identical to that of an ISJ and its SpliceAI score indicated a splicing gain, the exon triplet was considered with promoted exon inclusion under that corresponding mutation. If the coordinates of 5′ splice site of the middle exon identical to an ISJ and its SpliceAI score indicated a splicing loss, the exon triplet was considered as exon skipping under that corresponding mutation.

Three kinds of rescue were considered: (1) if a mutated exon triplet was predicted (by SpliceAI) to cause exon skipping and it was predicted (by our CNN model) to have an inclusion response after BPN-15477 treatment; (2) if a mutated exon triplet was predicated (by SpliceAI) to cause promoted exon inclusion and it was predicted (by our CNN model) to have an exclusion response after BPN-15477 treatment; and (3) if a mutated exon triplet generated a pre-mature termination codon (PTC) inside middle exon and it was predicted (by our CNN model) to have an exclusion response after BPN-15477 treatment and its reading frame was not shifted after skipping the middle exon.

**Allele frequency from gnomAD**. VCF files for both human exome and genome sequencing were downloaded from gnomAD (v2.1.1)[86,87]. The corresponding ClinVar mutations were located in these VCF files via their SNP IDs. If the short variant was found only in exome or only in genome sequencing VCF, the reported minor allele frequency was then used. If a short variant was found in both exome or in genome sequencing, the maximum value was taken.

**Statistical analyses**. For differential splicing analysis, Cochran–Mantel–Haenszel test was applied followed by FDR correction. An FDR < 0.1 and $\Delta\psi \geq 0.1$ was considered as inclusion-response after the treatment (Fig. 2b). Any triplet with FDR < 0.1 and $\Delta\psi \leq -0.1$ was considered as exclusion-response after the treatment (Fig. 2b). Any exon triplet, whose $\psi$ before-treatment range from 0.1 to 0.9 and $\psi$ change is <0.01, was considered as unchanged-response. For Pearson correlation (Fig. 2c), the gray zone indicates 95% confident intervals. For splicing strength comparison (Fig. 3c), the maximum entropy was compared among inclusion, exclusion, and unchanged groups at each splice junction, using two-tailed Welch's *t*-test. For the boxplots (Fig. 3c), the middle lines inside boxes indicate the medians. The lower and upper hinges correspond to the first and third quartiles. Each box extends to 1.5 times inter-quartile range (IQR) from upper and lower hinges, respectively. Outliers were not shown. For RT-PCR comparison (Figs. 3 and 4), two-tailed Welch's *t*-test was applied. In all plots, the error bars indicate ± standard deviation (SD) from mean values. In all plots, the significance levels were marked by *$p < 0.05$, **$p < 0.01$, and ***$p < 0.001$.

**Reporting summary**. Further information on research design is available in the Nature Research Reporting Summary linked to this article.

## Data availability

The datasets generated during and/or analyzed during the current study are available in the GEO database: GSE158947. Other public datasets used in this study: Human genome Ensembl GRCh37 [http://ftp.ensembl.org/pub/release-75/fasta/homo_sapiens/dna/]

Homo_sapiens.GRCh37.75.dna.primary_assembly.fa.gz], Human transcriptome Ensembl GRCh37.75 [http://ftp.ensembl.org/pub/release-75/gtf/homo_sapiens/Homo_sapiens.GRCh37.75.gtf.gz], ClinVar GRCh37 VCF version 20190325 [https://ftp.ncbi.nlm.nih.gov/pub/clinvar/vcf_GRCh37/archive_2.0/2019/clinvar_20190325.vcf.gz], and gnomAD v2.11 GRCh37 VCF [https://gnomad.broadinstitute.org/downloads#v2-variants]. Any other relevant data are available from the authors upon reasonable request. Source data are provided with this paper.

## Code availability

Source code to reproduce the results[88] in the paper is available on GitHub [https://github.com/talkowski-lab/SMC_CNN_Model].

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

## Acknowledgements
We thank the NIH Blueprint lead development team members, Dr. Charles Cywin, Dr. Lisa Minor, Dr. Ronald Franklin, Dr. Ronald White, Jamie Driscoll, Dr. Juan Marugan, Dr. Enrique Michelotti, Dr. Amir Tamiz, Dr. Bruce Molino, Dr. Douglas Kitchen, Keith Barnes, Dr. Katrina Gwinn, and Dr. Rebecca Farkas for their longstanding collaboration and helpful discussions. This work was supported by National Institutes of Health grants (U01NS078025, R21NS095437, R01NS102423, and R37NS095640 to S.A.S. and M.E.T.). Research support from PTC Therapeutics, Inc. (S.A.S.).

## Author contributions
D.G., E.M., M.S., M.E.T., and S.A.S. conceived and designed the research. D.G. performed all of the in silico analyses and built the CNN model for potential therapeutic target prediction. E.M., M.S., A.J.K., E.M.L., Y.Y., N.S., and X.Z. performed experiments. W.L. performed the *K*-mer enrichment analysis. D.G., E.M., and M.S. analyzed data. G.J., W.P., A.R., S.E., A.C., A.D., N.A.N., C.T., K.A.E., M.W., G.K., and J.N. provided advice and participated in the medicinal chemistry. D.G., E.M., M.S., M.E.T., and S.A.S. wrote the original manuscript, which was reviewed and edited by all authors. G.R.C., M.E.T., and S.A.S. supervised the research.

## Competing interests
W.L., A.D., N.A.N., C.T., K.E., M.W., G.K., Y.Y., V.G., J.N., and X.Z. are employees of PTC Therapeutics, Inc., a biotechnology company. In connection with such employment, the authors receive salary, benefits and stock-based compensation, including stock options, restricted stock, other stock-related grants, and the right to purchase discounted stock through PTC's employee stock purchase plan. Personal financial interests: S.A.S. is a paid consultant to PTC Therapeutics and is an inventor on several U.S. and foreign patents and patent applications assigned to the Massachusetts General Hospital, including U.S. Patents 8,729,025 and 9,265,766, both entitled "Methods for altering mRNA splicing and treating familial dysautonomia by administering benzyladenine," filed on August 31, 2012 and May 19, 2014 and related to use of kinetin. S.A.S., G.J., and W.D.P. are inventors on U.S. Patent 10,675,475, assigned to Massachusetts General Hospital and entitled "Compounds for improving mRNA splicing" filed on July 14, 2017 and related to use of BPN-15477. D.G., E.M., W.L., K.A.E., C.R.T., Y.Y., V.G., A.D., N.A.N., M.E.T., and S.A.S. are inventors on an International Patent Application Number PCT/US2021/012103, assigned to Massachusetts General Hospital and PTC Therapeutics entitled "RNA Splicing Modulation" related to use of BPN-15477 in modulating splicing. All other authors declare no competing interests.
