## [Peer Review File · Nature Communications]

Reviewers' Comments:

Reviewer #1:

Remarks to the Author:

The authors present a novel compound, BPN-15477, that promotes exon inclusion. The authors are motivated by the IVS20+6T>C mutation, which is known to increase skipping of exon 20 of the Elongator complex protein 1 (ELP1) gene and causes familial dysautonomia (FD). They show that BPN-15477 fully rescues splicing of ELP1 in cell lines from FD patients at reasonable concentrations, making it a potential therapeutic candidate.

To understand the mechanisms of BPN-15477 and to further explore its therapeutic potential, the authors treat wildtype (WT) cell lines with BPN-15477 and record the splicing changes. Then they train a convolutional neural network (CNN) to learn to predict which exon triplets will have increased middle-exon inclusion or exclusion after BPN-15477 treatment. They use this model to make predictions at ClinVar pathogenic and likely-pathogenic mutations that are also predicted to affect splicing by SpliceAI and they identify hundreds of such mutations that could potentially be targeted by BPN-15477. Using patient cell lines or minigene constructs, they validate predictions in disease-causing genes for cystic fibrosis (CFTR), cholesterol ester storage disease (LIPA), Lynch syndrome (MLH1) and familial frontotemporal dementia (MAPT), showing that BPN-15477 treatment has indeed the predicted effect on exon inclusion/exclusion.

Although the results shown for ELP1 splicing rescue suggest a strong therapeutic potential for BPN-15477 and the performance of the predictive model for finding additional therapeutic targets is promising, I believe that the analyses of the mechanism of action and sequence specificity of BPN-15477 need to be strengthened. The authors should:

- 1) improve the motif analysis of the CNN, which would shed light into the mechanism of action of BPN-15477,
- 2) address the question of how BPN-15477, which was selected to increase exon inclusion in ELP1, leads to increased exon skipping in many genes,
- 3) address concerns regarding the SpliceAI analysis.

More detailed suggestions in these areas are below.

1) Motif analysis and reproducibility of the CNN

The authors train a CNN to predict whether an exon triplet would have increased inclusion or exclusion of the middle exon after BPN-15477 treatment, based on the sequences around the splice sites of the triplet. Then they attempt to extract motifs that explain the behavior of the CNN. However, the motif contribution and positional motif contribution results are hard to interpret. A lot of the motifs and positional contributions that they show could be artifacts of the baselines that they use for comparison (eg. average activation, which might correspond to motifs never present in real sequences) or simply non-reproducible neural-network noise.

In particular, some of the results are counter-intuitive. Eg. the authors claim that motif09 is important at positions -1 and +1 of the 5'ss. But then the motif should have at least a weak GT within its first 3 positions, which is not the case.

Below are a few suggestions for improving interpretation:

- a) Show examples of sequences where each first layer filter has high activations.
- b) Use a method like DeepLift, input*gradients, or in-silico mutagenesis of every input base, to show which bases contribute to the prediction made for specific examples. Of particular interest for understanding the underlying mechanisms would be examples where the model predicts increased

exon exclusion.

c) For positional importance, show which positions lead to the highest activations of each filter (instead of comparing to an artifactual baseline).

d) For the motif enrichment analysis (Sup Fig 3b), the authors should compare the model motifs/filters to the motifs present at the same positions in the input sequences. Eg. in Sup Fig 3b, the AGTAA motif captures the donor motif starting at position -1. The "most similar" motif49 is presumably enriched at a different position (-2) so the two logos are not capturing the same thing. In the same figure, motif29 doesn't look similar at all to the LOGO above it.

e) The authors should also address the question of the reproducibility of their results. Their CNN has several orders of magnitude more parameters than training examples. NNs can be very volatile especially with so few training examples. Why did the authors choose a model with that many parameters? Are the discovered motifs reproducible across random initializations of the model?

2) How does BPN-15477 promote exon skipping?

Although BPN-15477 was initially discovered in an effort to promote exon inclusion in ELP1, in WT fibroblasts, the authors report more than twice as many cases of increased exon skipping compared to increased exon inclusion. The authors suggest that BPN-15477 acts by promoting recruitment of U1 snRNP at non-canonical 5' splice sites. Some questions that the authors should consider are:

a) Do the exons whose skipping is increased tend to have suboptimal 5' splice sites that get activated by BPN-15477? The improved motif analysis as suggested above should hopefully help clarify this mechanism.

b) Does BPN-15477 treatment have other effects besides changing the rate of exon inclusion or skipping? For example, increased levels of intron retention or increased usage of alternative/suboptimal splice sites might not be captured by the measured percent splice in.

3) SpliceAI analysis

The authors might have misunderstood some of the SpliceAI parameters and results. By default, SpliceAI predicts how a variant changes splicing patterns within 50bp of the variant. To make this determination, SpliceAI uses +/-5Kb of context around the variant. Thus, the predicted effects will never be more than 50bp away from the variant under consideration. The predicted effects can be the loss/weakening or gain/strengthening of a splice acceptor or donor. SpliceAI outputs both the type and the position of the effect. The authors claim that "~20% of all CV-pMUTs are predicted to alter splicing within 5 kb of the mutation" (L168). This should be 50bp, not 5 kb.

Similarly they say that ~98% of ClinVar mutations predicted to disrupt annotated splice sites were within 75bp of the splice site (L171). However, by design 100% of such mutations should be within 50bp of the splice site that is predicted to be disrupted. The predicted disrupted splice site is not necessarily an Ensembl-annotated site, which is probably where the discrepancy is coming from. The authors should consider the site that SpliceAI is predicting to be disrupted and not the annotated site nearest to the variant. If the two sites differ, the output SpliceAI score refers to the former and does not apply to the latter. In particular, in the "Rescue definition and prediction" section, the authors should make sure that the affected junction predicted by SpliceAI (ISJ) is identical with the junction considered in their analysis, not simply overlapping the same domain. For example, the authors consider exon skipping events "if the 5' splice site of the middle exon overlapped with an ISJ and its SpliceAI score indicated a splicing loss". If the ISJ was not the same base as the annotated 5' splice site of the middle exon, then SpliceAI is predicting a weakening of

a DIFFERENT splice site than the annotated splice site.

Other comments/suggestions:

4) In some of the experimental validations, it's unclear whether the authors made predictions on the WT or the mutant sequence. Eg. in section "Identification of potential therapeutic targets of BPN-15477" did they input the WT or the mutant allele of the considered 11,616 exon triplets? Similarly in the experimental validation section they claim: "As predicted, treatment with BPN-15477 led to a significant reduction of exon 10 inclusion in the wild type MAPT transcript." However, they say in line 216 that they made predictions for the mutated sequence, not for the WT.

5) It would be very interesting to study potential off-target or other unintended effects of BPN-15477. In WT cell lines, are there any global changes in expression levels or cell viability? Similarly, since different exon triplets are normally expressed in different tissues, it'd be interesting to use the CNN to predict which/how many expressed exon triplets would be affected across different tissues (using for example GTEx data to get the expressed exons).

Minor comments:

6) Why did they only use a few hundred (out of several thousand) unchanged exon triplets to train their CNN?

7) Why was it necessary to standardize the output scores as described in section "Standardized probability from prediction"? This isn't common practice.

8) In section "Candidate selection for minigene validation" what do the authors mean by condition 3)?

9) For computing gnomAD allele frequencies, it'd be better to take the average or maximum of exon and genome allele frequencies. Their current computation is a little problematic, since the samples used for exome and genome sequencing (and therefore also the observed counts) are not independent.

Regards,

Sofia Kyriazopoulou Panagiotopoulou

Reviewer #2:

Remarks to the Author:

Slaugenhaupt and co-workers identify the splicing modulator BPN-15477 for familial dysautonomia and go on to evaluate its in vitro potential to correct splicing targets in a range of other human genetic disorders including in CFTR, LIPA, MLH1 and MAPT, all identified via a deep learning approach. While interesting and generally well-written, the manuscript as it stands raises fundamental questions which in this reviewer's opinion preclude publication at the present time.

Major comments / questions

1. Confusing as to why SMN2 was not studied given history with risdiplam.
2. BPN-15477: no mechanistic data for drug action provided – this should be directly addressed.
3. Effects of BPN-15477 in validation experiments are marginal and not very impressive. It is not convincing that these would translate into meaningful changes in protein expression (although this is argued for the case of LIPA, without evidence). Evidence should be provided and in a paper such as this, reporting a novel methodology, at least one target should be studied in an appropriate in vivo model to provide direct evidence of likely disease benefit.
4. Validation experiments are primarily based on artificial minigene systems with the exception of mutant fibroblasts in the case of LIMA and ELP1. No endogenous protein expression data is

provided – this should be directly addressed.

5. All assessments of splicing in validation experiments are assessed at the RNA level by semi-quantitative RT-PCR. Data from quantitative PCR methods such as ddPCR or RT-qPCR should be provided. Figure 3d data for mutated constructs are particularly unconvincing.

6. The authors identify specific motifs which explain ~95% of their model performance. It is essential that these motifs are independently validated by an orthogonal method e.g. an ASO which specifically targets those regions.

7. The paper draws parallels to ASO targetable exon-skipping diseases. However, the biggest drawback of BPN-15477 is the observation of unspecific splicing modulation (934 exon triples show significant modulation of the middle exon). The effects of this unspecific splicing alteration are not discussed but are certainly of concern from a therapeutic aspect. A more detailed analysis (ideally in vivo) and discussion of the limitations of the current approach are required.

8. Ironically, it would be difficult to train a model like this with a drug which doesn't have unspecific effects, because you would not have enough data to train the model. This limits the potential applicability of this approach for drugs with better specificity and appears to be a major limitation of the current work.

Minor Points

1. The apostrophe symbol is incorrectly used instead of the prime symbol throughout

2. In Figure 1, P1 and P2 are not described. These are presumably the primers used for RT-PCR in other figures.

3. I assume that 'Normalized RLU' in Figure 1 refers to the FLuc:RLuc ratio. If true, this should be clarified.

4. The percent spliced index (psi) is confusingly used throughout. It seems that psi is a ratio and not a percentage, so the nomenclature is inaccurate

5. Figure 2a is confusing. If I understand correctly R1, R2 and R3 are supposed to represent reads, but this is not clear from the diagram. At present these labels appear to show the intron or intron-exon-intron sequence that is spliced out.

Reviewer #3:

Remarks to the Author:

In this manuscript, authors present the development of a compound that corrects splicing of exon 20 of ELP1 gene carrying the major IVS20+6T>C mutation which is responsible for familial dysautonomia. Further, they applied machine learning approaches to evaluate the therapeutic potential of this compound to correct splicing in other human genetic diseases. This manuscript is a good example of how deep learning techniques can identify sequence signatures and predict response to pharmacological modulation. I am not an expert on machine learning so I will circumscribe my review comments to the aspects related with splicing. I assume that other reviewer (s) will comment on the machine learning aspects of the manuscript.

Major

- Figs. 4c-f: really the changes observed in the mutated lines with the BPN-15477 compound are very small (3-10% change). How much of that little change is caused by the compound or any other secondary effect?
- Related with the previous point: how significant is the change in response to BPN-15477 in terms of physiologic relevance? I think this is something important to have this manuscript published in Nature Communications and it will for sure make the paper much stronger combining: splicing correction with a small molecule + physiological impact of that correction + machine learning.
- Please, indicate the number of independent experiments in each figure legend. It should be at least three.

Minor

1. SMC abbreviation in the abstract is not defined.
2. Avoid green red combinations (color blind readers).

Reviewer #4:

Remarks to the Author:

Gao et al., present a deep-learning approach to predict splicing events that will be affected by the splicing modulator compound, BPN-15477, a more potent and efficacious derivative of the natural product, kinetin, in vitro. Kinetin was shown previously by the authors to promote exon 20 inclusion in the ELP1 IVS20+6T>C mutation associate with familial dysautonomia (FD). In this manuscript they used a deep convolutional neural network to identify additional splicing targets of the new compound and validated a number of these to demonstrate the utility of the approach. With splicing-related defects being a prevalent cause of disease and as more splicing modulator compounds are being discovered, tools such as the one presented in this manuscript are valuable for understanding and predicting potentially harmful off-target effects as well as for identifying additional therapeutic targets for the molecules. The molecule tested here is similar to another kinetin derivative reported by Yoshida et al., in 2015. In that study, similar analyses to the current manuscript are presented and motifs are also identified that appear to be common to compound-responsive splicing events. While the current study by Gao et al., offers a more sophisticated approach to identifying compound targets via deep machine learning, the overall experimental approach and molecule do not offer a major advance in our understanding of this class of drugs. Nonetheless, the study is a good proof-of-concept that presents a potentially valuable approach for identifying drug targets for this and other splice modulating compounds. The study also identifies some interesting possible therapeutic targets for BPN-15477, though the high concentrations of the compound used to identify splicing targets (30 micromolar) may limit the therapeutic value of this specific molecule. There are a number of issues with the reporting in the manuscript (both minor and major) that should be addressed as detailed below.

1. Figure 1: b) should include EC50 data for both BPN-15477 and Kinetin. C) splicing correction is shown for the purpose of validating splicing correction by RT-PCR. The low end of the dose curve is missing as is the comparison to kinetin.
2. For the RNA-seq samples, were the cells treated a single time with the compound and then incubated for seven days? What is known about the stability of the compound in the culture media? Why was this long incubation time selected? Fig 1b and c show results from fibroblast cells treated for only 24 hours.
3. Provide citation for statement: "This is exciting since a small increase in functional CFTR protein is associated with a significant improvement in patient phenotype".
4. Provide raw data for all data points in Fig. 2C. This could be in the form of a Table that lists the name of the genes, the exon # that is changing and the PSI values.
5. In terms of selectivity, as presented on page 5 line 116, it would be informative to also include the number of alternatively spliced exons (triplets with two isoforms) that do not change. This number, I presume is the same as the unchanged-response set (382) exon triplets with two expressed isoforms described on p. 6 line 129.
6. In the bar graphs for the drug response plotting standardized probability, the values for the y-axis are not completely labeled with the bars extending beyond the y-axis labels, making it difficult to estimate the actual value.
7. Fig. 3 and 4. These motifs are very short and degenerate, it is surprising that they could have

any predictive value as they will be expected to be very frequent. Is there a way to score the motif match for each of these?

8. The figure legend of Supp. Fig 3 indicates that the RT-PCR was performed in triplicate to make the bar plot. This use of technical replicates is not as robust as the use of experimental replicates and lessens the strength of the results. Similarly in Fig. 3 d-e, the legend states that each RT-PCR was done in duplicate and independently repeated three times for each minigene, does this mean that the same RNA sample from a single transfection/treatment was analyzed six times or was the transfection/treatment performed three times? Please clarify.

9. Fig. 4c,d it would be helpful to indicate the position of the mutation and the effect on the motif.

10. The discussion could be expanded to consider in more detail the meaning of the results and how they relate to previous studies and the future of the molecule as a therapeutic. For example, the authors should comment on whether the motifs represent direct targets of the compound or just identify features of "weak" or "strong" splice sites that might be sensitive to subtle alterations in splicing in general induced by the compound. Further, it is a big leap to conclude from the motif analysis associated with sensitive exon 5'ss that the result is consistent with this class of compounds promoting the recruitment of U1 snRNP to non-canonical 5' splice sites. They cite a study by Yoshida et al., which shows that a kinetin-like molecule promotes U1-binding to the weak 5'ss of ELP1. However, Yoshida et al., actually go on to identify cis-acting sequence elements common to splicing events responsive to compound treatment and conclude that sequence elements that appear to act as hnRNP H binding exonic splicing silencers and may be the target of the compounds and that inactivating or interfering with these ESSs may allow more effective U1 snRNP binding simply because splicing is derepressed by the ESS.

11. Regarding Yoshida et al. further, the authors should discuss these previous results in more detail, perhaps commenting on the similarities or differences in identified motifs as well as the identified responsive splicing events.

Reviewer #1

Although the results shown for ELP1 splicing rescue suggest a strong therapeutic potential for BPN-15477 and the performance of the predictive model for finding additional therapeutic targets is promising, I believe that the analyses of the mechanism of action and sequence specificity of BPN-15477 need to be strengthened.

Major comments:

1) Motif analysis and reproducibility of the CNN.

The authors train a CNN to predict whether an exon triplet would have increased inclusion or exclusion of the middle exon after BPN-15477 treatment, based on the sequences around the splice sites of the triplet. Then they attempt to extract motifs that explain the behavior of the CNN. However, the motif contribution and positional motif contribution results are hard to interpret. A lot of the motifs and positional contributions that they show could be artifacts of the baselines that they use for comparison (eg. average activation, which might correspond to motifs never present in real sequences) or simply non-reproducible neural-network noise.

We appreciate the reviewer's insight regarding the reproducibility of our model and the approach of motifs analysis. To address these concerns, our revision includes the following:

I) To assess reproducibility, we retrained the model 1,000 times using random initialization. We then tested the correlation between the motifs derived from 1,000 random initiated models and those derived from the original model, by which we found our CNN model trained from a relatively small sample size was stable and robust (Supplementary Fig. 2d-e).

II). For motif positional importance analysis, we investigated the highest activation of each position in the first layer, instead of comparing to an artificial baseline (Fig. 3b). The new result is consistent with the original analysis, emphasizing the importance of sequence patterns around the 5' splice site.

III) To assign each motif to the treatment-response class it contributes most to, we have investigated AUC changes, which are more informative than activation information alone. To avoid comparing to an artificial baseline, now we zeroed the activation from each motif in the first layer, as if the motif was not initially detected, and then calculated the AUC changes (Fig. 3a). We found thirteen motifs explained 92.62% of the AUC, nine of which were already reported among the twelve most significant motifs in our initial submission. We believe these analyses have greatly improved our interpretation of the results.

In particular, some of the results are counter-intuitive. Eg. the authors claim that motif09 is important at positions -1 and +1 of the 5'ss. But then the motif should have at least a weak GT within its first 3 positions, which is not the case.

This observation is further explicated in our updated motif positional importance using activation of the first layer according to the reviewer's suggestion (Fig. 3b). The Motif09 ([T/C]TTG[A/T]) now shows its highest activation at the -3 position. Its fourth and fifth positions are G[A/T], which aligns with +1 and +2 positions of the 5' splice site. It is also important to note that the new positional importance estimated by activation is identical to the initial positional importance estimated by entropy changes for almost all motifs (Fig. 3b also see answers to Q1c).

a) Show examples of sequences where each first layer filter has high activations.

Figure R1

The heatmap above (Figure R1) represents the average activation per sequence from the test set (by column) of each motif (by row), extracted from the first layer of our CNN model. The sequences with inclusion, exclusion and unchanged responses detected from RNASeq are grouped together and marked by red, blue and black on the top X-axis of the heatmap. Each motif is marked by either red, blue or black depending on which class prediction it contributes most to, according to the motif importance analysis (Fig. 3a, Supplementary Fig. 3). We have the following observations: a) most motifs derived from our CNN model have medium-to-high overall activations in the first layer; b) our top 13 motifs in Fig. 3a all have medium-to-high average activations; c) The summarized activation per motif (e.g. by taking average as shown in the heatmap above, maximum, or 75 percentile per motif) is not sufficient to distinguish treatment response. Also see the answers to Q1b and Q1c below.

b) Use a method like DeepLift, input*gradients, or in-silico mutagenesis of every input base, to show which bases contribute to the prediction made for specific examples. Of particular interest for understanding the underlying mechanisms would be examples where the model predicts increased exon exclusion.

We appreciate the reviewer's suggestion to improve the CNN-derived motif analysis. We have implemented *in silico* saturation mutagenesis (Supplementary Fig. 4). This result revealed that the base contribution to the treatment response was consistent with our positional analysis in Fig. 3b and peaked in proximity to the 5' splice sites of the middle exons. It also demonstrated distinct activation patterns among sequences

with inclusion, exclusion and unchanged responses, although it is not a direct interpretation of the molecular mechanism of the treatment.

- c) *For positional importance, show which positions lead to the highest activations of each filter (instead of comparing to an artifactual baseline).*

We appreciate the reviewer's suggestion. We re-applied the analysis as suggested (Fig. 3b), and found comparable results to our initial submission, highlighting the significance of the motif importance around the 5' splice site, and that our analyses are robust. Notably, the most important positions for each motif are almost the same between the two analyses.

- d) *For the motif enrichment analysis (Sup Fig 3b), the authors should compare the model motifs/filters to the motifs present at the same positions in the input sequences. Eg. in Sup Fig 3b, the AGTAA motif captures the donor motif starting at position -1. The "most similar" motif49 is presumably enriched at a different position (-2) so the two logos are not capturing the same thing. In the same figure, motif29 doesn't look similar at all to the LOGO above it.*

We did not initially perform a quantitative analysis of the similarities between the derived CNN and the *k*-mer-enrichment-derived motifs, and we thank the reviewer for this suggestion. In the revised manuscript, we have applied a Pearson correlation to reveal the most similar matches between the two sets, and these results are now provided in Supplementary Fig. 5a.

- e) *The authors should also address the question of the reproducibility of their results. Their CNN has several orders of magnitude more parameters than training examples. NNs can be very volatile especially with so few training examples. Why did the authors choose a model with that many parameters? Are the discovered motifs reproducible across random initializations of the model?*

We appreciated the reviewer's comment on our CNN model stability and reproducibility. We have now implemented an L1 regularization and dropout approach to control overfitting in the original model. According to the reviewer's suggestion, we also used the following approaches to demonstrate the robustness of our original model: we retrained the model 1,000 times using random initialization and investigated the AUC distribution across all these models for each class (Supplementary Figure 2d). We also found a high correlation between top 10 motifs derived from random-initiated models and our original CNN model (Supplementary Fig. 2e). Collectively, these above analyses have improved the robustness and nuance of our interpretation, but none of our additional analyses have altered our initial conclusions.

2) *How BPN-15477 leads to increased exon skipping in many genes.*

Although BPN-15477 was initially discovered in an effort to promote exon inclusion in ELP1, in WT fibroblasts, the authors report more than twice as many cases of increased exon skipping compared to increased exon inclusion. The authors suggest that BPN-15477 acts by promoting recruitment of U1 snRNP at non-canonical 5' splice sites. Some questions that the authors should consider are:

- a) *Do the exons whose skipping is increased tend to have suboptimal 5' splice sites that get activated by BPN-15477? The improved motif analysis as suggested above should hopefully help clarify this mechanism.*

We agree that the discovery of exon skipping following treatment is intriguing, and a study of the mechanism of drug action is certainly warranted in the future. We did investigate suboptimal 5' splice sites as the reviewer suggested by calculating suboptimal splice site usage as a ratio relative to the expression of its nearby optimal splice site. Among 18,639 suboptimal splice sites detected in our RNASeq data, we found only one of them with decreased usage against the treatment (Fig. R2a below), suggesting exon skipping caused by the drug was not due to the enrichment of suboptimal 5' splice sites. We did, however, observe that the skipped exons preferred T at the last nucleotide position in the new analysis using *in silico* saturation mutagenesis (Supplementary Fig. S4).

Figure R2

- b) *Does BPN-15477 treatment have other effects besides changing the rate of exon inclusion or skipping? For example, increased levels of intron retention or increased usage of alternative/suboptimal splice sites might not be captured by the measured percent splice in.*

We appreciate the reviewer's suggestion to comprehensively examine whether any other types of splicing events were altered by the treatment and agree that this is a logical next step for our future studies. We did calculate the IR ratio using a published algorithm, IRFinder (<https://github.com/williamritchie/IRFinder>). We detected 99 (0.04%) differential IR events among 236,293 annotated introns (Fig. R2b, red dots indicate significant IR changes). We focused this work on the study of intron skipping/retention by studying exon triplets, but again agree that it will be important to look at other splicing changes in the future.

- 3) SpliceAI analysis.

The authors might have misunderstood some of the SpliceAI parameters and results. By default, SpliceAI predicts how a variant changes splicing patterns within 50bp of the variant. To make this determination, SpliceAI uses +/-5Kb of context around the variant.

Thus, the predicted effects will never be more than 50bp away from the variant under consideration. The predicted effects can be the loss/weakening or gain/strengthening of a splice acceptor or donor. SpliceAI outputs both the type and the position of the effect. The authors claim that “~20% of all CV-pMUTs are predicted to alter splicing within 5 kb of the mutation” (L168). This should be 50bp, not 5 kb. Similarly they say that ~98% of ClinVar mutations predicted to disrupt annotated splice sites were within 75bp of the splice site (L171). However, by design 100% of such mutations should be within 50bp of the splice site that is predicted to be disrupted. The predicted disrupted splice site is not necessarily an Ensembl-annotated site, which is probably where the discrepancy is coming from.

We regret that we were not sufficiently clear on this issue and have updated the main text to reflect that SpliceAI predicts the splice sites with potentially altered splicing within 50bp of the mutations. We have also removed the original Fig. 4b, which is now unnecessary, and Fig 4a now indicates that the predicted splice sites disrupted are not necessarily Ensembl-annotated sites. Please note that this did not change our initial results for potential BPN-15477 targets.

The authors should consider the site that SpliceAI is predicting to be disrupted and not the annotated site nearest to the variant. If the two sites differ, the output SpliceAI score refers to the former and does not apply to the latter. In particular, in the “Rescue definition and prediction” section, the authors should make sure that the affected junction predicted by SpliceAI (ISJ) is identical with the junction considered in their analysis, not simply overlapping the same domain. For example, the authors consider exon skipping events “if the 5’ splice site of the middle exon overlapped with an ISJ and its SpliceAI score indicated a splicing loss”. If the ISJ was not the same base as the annotated 5’ splice site of the middle exon, then SpliceAI is predicting a weakening of a DIFFERENT splice site than the annotated splice site.

We appreciate the reviewer’s attention to the details behind our approach. In our original report, the word “overlap” was intended to mean “identical” (as we were comparing a single nucleotide position predicted by SpliceAI to another single nucleotide position named ISJ). We apologize for the confusion and have made this change in the main text.

Other Suggestions/Comments:

- 4) *It’s unclear whether the authors made predictions on the WT or the mutant sequence.*

For model validation, we made predictions on both WT and mutant sequences. For the potential drug target exploration, we predicted the mutant sequence only. We now have updated the main text to make this clear.

- 5) *It would be very interesting to study potential off-target or other unintended effects of BPN-15477. In WT cell lines, are there any global changes in expression levels or cell viability? Similarly, since different exon triplets are normally expressed in different tissues, it’d be interesting to use the CNN to predict which/how many expressed exon triplets would be affected across different tissues (using for example GTEx data to get the expressed exons).*

Similar to previously reported splicing modulator compounds (Naryshkin et al. Science, 2014; Palacino et al. Nat. Chem. Biol. 2015; Woll et al. RNA Therapeutics, 2018). BPN-

15477 treatment had minimal effect on global gene expression; only 557 of 17812 (3.24%) genes were differentially expressed in the BPN-15477 treated cells compared with the vehicle-treated cells. Further, we performed *in vivo* experiments in transgenic animals with no negative health outcomes. We appreciate the reviewer's suggestion to investigate the tissue specific effect of the treatment on gene expression, which would be very interesting, but the goal of this study was to identify new targets for this class of splicing modulator compounds.

Minor Comments:

- 6) *Why did they only use a few hundred (out of several thousand) unchanged exon triplets to train their CNN?*

We chose to only use exon triplets for which we had high confidence to determine that splicing was unchanged. Thus, we decided to restrict to those triplets whose initial PSIs in WT were between 0.1 to 0.9 and PSI changes after treatment were less than 0.01, which we would consider as high-confidence unchanged triplets.

- 7) *Why was it necessary to standardize the output scores as described in section "Standardized probability from prediction"? This isn't common practice.*

We appreciate the opportunity to further describe this approach. There are several components to implementing this standardization. First, the training data is not balanced, so we adjusted the classification threshold against the training size of each class and set a 95%-specificity threshold on the ROC curve of each class. Second, the use of such thresholds made a direct comparison across prediction values (probabilities) of the three classes challenging in a bar plot format. Thus, we re-calculated post-prediction adjusted probabilities relative to the threshold of each class. Finally, the adjusted probabilities were scaled for the purposes of visualization of all prediction results after adjustment on the same scale. Most importantly, this standardization process does not change the conclusion from the CNN prediction compared to non-standardized results.

- 8) *In section "Candidate selection for minigene validation" what do the authors mean by condition 3)?*

The third condition means that the minigenes selected for validation must recapitulate the splicing pattern of the endogenous gene.

- 9) *For computing gnomAD allele frequencies, it'd be better to take the average or maximum of exon and genome allele frequencies. Their current computation is a little problematic, since the samples used for exome and genome sequencing (and therefore also the observed counts) are not independent.*

We appreciate the reviewer's suggestion. We have changed to use the maximal allele frequencies between exon and genome sequencing data. We have updated the corresponding results in the main text, main table and supplementary tables.

Reviewer #2

Slaugenhaupt and co-workers identify the splicing modulator BPN-15477 for familial dysautonomia and go on to evaluate its in vitro potential to correct splicing targets in a range of other human genetic disorders including in CFTR, LIPA, MLH1 and MAPT, all identified via a

deep learning approach. While **interesting and generally well-written**, the manuscript as it stands raises fundamental questions which in this reviewer's opinion preclude publication at the present time.

Major comments / questions:

1) *Confusing as to why SMN2 was not studied given history with risdiplam.*

We previously established that this class of SMCs do not affect the splicing of SMN2 exon 7. Kinetin, the original lead compound, was identified while participating in the NINDS Sponsored Drug Screening Consortium and teams studying SMA participated in that consortium (Hims et al. J Mol Med, 2007, Heemskerk et al. Trends Neurosci, 2002). Likewise, small molecules that alter splicing of SMN2 (aclarubicin, valproic acid, risdiplam) do not correct splicing of ELP1 exon 20. Further, the RNASeq results from the current study confirmed that BPN-15477 does not change the splicing of SMN2.

2) *BPN-15477: no mechanistic data for drug action provided.*

The main objective of the current work was to evaluate the effect of BPN-15477, a more potent analog of kinetin, on splicing and to use it to build and validate our machine learning model with the goal of identifying new potential targets for therapy. Our previous work and that of others suggests that this class of SMCs are likely promoting recruitment of U1 to the splice site, either directly or indirectly. We have modified the manuscript accordingly.

3) *Effects of BPN-15477 in validation experiments are marginal and not very impressive. It is not convincing that these would translate into meaningful changes in protein expression (although this is argued for the case of LIPA, without evidence). Evidence should be provided and in a paper such as this, reporting a novel methodology, at least one target should be studied in an appropriate in vivo model to provide direct evidence of likely disease benefit.*

We thank the reviewer for raising this critically important and frequently discussed misconception that the magnitude of in vitro effect size can necessarily be translated into predictions of clinical efficacy. In FD, we have previously demonstrated and published that a 10% increase in functional protein can have a dramatic phenotypic effect (Morini et al. AJHG 2019 and Dietrich et al. HMG 2012), and this is true for many other human recessive diseases. Several studies have shown that CF protein levels as low as 10% of WT result in improved lung function and survival, and protein levels of as low as 3% of WT can rescue pancreatic insufficiency, even in moderate to advanced stages of disease (McCague et al. Am J Respir Crit Care Med 2019; Ramsey et al. N Engl J Med 2011; Wainwright et al. N Engl J Med 2015; Taylor-Cousar et al. N Engl J Med 2017; Davies JC et al. N Engl J Med 2018; Keating et al. N Engl J Med 2018). Another good example is represented by the small increase in lysosomal acid lipase (LAL or LIPA) residual enzyme activity. Wolman disease and cholesterol ester storage disease (CESD) are both caused by mutations in the LIPA gene. Wolman is lethal in infancy, whereas CESD patients with only 3% of the normal level of LIPA transcript have a much milder clinical course (Aslanidis et al. Genomics 1996). Duchenne muscular dystrophy (DMD) is a severe neuromuscular disorder caused by frameshift mutations that completely eliminate the production of dystrophin. Several therapeutic approaches for DMD, including the antisense oligonucleotide eteplirsen for DMD exon 51 skipping and golodirsen and viltolarsen for DMD exon 53 skipping, are evolving with the aim to transform the severe DMD into the milder Becker form of the disease by only

partially restoring the expression of a mutated or truncated dystrophin. Viltolarsen raises dystrophin production by about 5%, yet it can stabilize and improve muscle strength and functionality (Dhillon S. Drugs 2020, Clemens et al. JAMA Neurol. 2020 and FDA Approves Targeted Treatment for Rare Duchenne Muscular Dystrophy Mutation. 2020). Taken together, all of these studies show that even very small increases in the level of functional protein can have a dramatic impact on phenotype.

As suggested by the reviewer, we have now included new *in vivo* data of BPN-15477 treatment in our humanized transgenic mouse model. We show that BPN-15477 treatment leads to *ELP1* splicing correction and increases *ELP1* protein in all tissues examined. In addition, in this revised version of the manuscript we have included new data showing that splicing correction using BPN-15477 leads to increased protein production in disease relevant cellular models of LIPA and CFTR. Lastly, we have also included new data generated in collaboration with Dr. Gary Cutting's laboratory showing that BPN-15477 significantly increases CFTR chloride channel function in bronchial epithelial (CFBE) cell lines.

This new data clearly shows that the observed magnitude of splicing correction translates into a meaningful increase in protein levels.

4) *Validation experiments are primarily based on artificial minigene systems with the exception of mutant fibroblasts in the case of LIPA and ELP1. – this should be directly addressed.*

Expression minigenes (EMG) are the most common systems used in the field to study splicing. The minigenes used in our assays have been well-studied and accurately model the mis-splicing seen in patient cell lines. We have now added new mouse and protein validation data as described above which clearly shows that the minigene assays are representative and accurate.

5) *All assessments of splicing in validation experiments are assessed at the RNA level by semi-quantitative RT-PCR. Data from quantitative PCR methods such as ddPCR or RT-qPCR should be provided. Figure 3d data for mutated constructs are particularly unconvincing.*

Our long-standing experience in quantifying splicing isoforms validates the reliability and accuracy of assessing relative splicing changes using semi-quantitative RT-PCR as described (Cuajungco et al. 2003, Hims et al. J Mol Med, 2007, Shetty et al. HMG 2011; Donadon et al. HMG 2018 and Morini et al. AJHG 2019). It has proven difficult to amplify and correctly quantify with high specificity single isoforms due to the high sequence similarity using ddPCR or RT-qPCR. A direct comparison of RT-PCR vs RNAseq, confirming the accuracy of the method, is now shown in Supplementary Table 3.

6) *The authors identify specific motifs which explain ~95% of their model performance. It is essential that these motifs are independently validated by an orthogonal method e.g. an ASO which specifically targets those regions.*

The experimental validations included in the manuscript demonstrate the accuracy of our motif-based predictions. It is also important to note that the treatment response is not determined by any of these motifs independently, which is the strength of this approach, and therefore direct validation of a specific motif is not warranted. The splicing changes are quite specific, and the addition of the new functional data showing increases in *ELP1*, *LIPA*,

and CFTR protein offer further validation of the predictive power of our method and the specificity of action of our compound.

7) The paper draws parallels to ASO targetable exon-skipping diseases. However, the biggest drawback of BPN-15477 is the observation of unspecific splicing modulation (934 exon triples show significant modulation of the middle exon). The effects of this unspecific splicing alteration are not discussed but are certainly of concern from a therapeutic aspect. A more detailed analysis (ideally in vivo) and discussion of the limitations of the current approach are required.

Although we consider our compound quite specific since we observed splicing changes in only 0.58% of all expressed triplets (934 out of 161,097 expressed triplets), we would like to highlight that we used a low stringency cutoff in order to increase the number of splicing changes used in the machine learning model and to increase the power of prediction ($|\Delta\psi| \geq 0.1$ and $FDR < 0.1$). If we increased the stringency of the threshold, for example, fewer exons would be affected. We understand that specificity is an important consideration and we recognize that our compound needs additional medicinal chemistry optimization for any specific indication in order to increase the potency and decrease the off-target effects. Also, whether off-target effects observed in cells in a dish will lead to any undesired consequences to an organism is unclear. It should be noted that kinetin, which is less potent and a more promiscuous splicing modifier than BPN-15477, has been used in preclinical studies in mice with treatment lasting well over 2 years with no negative health consequences. Likewise, BPN-15477 was tested in vivo as reported in this manuscript and had no gross toxicity to the mice. Eventually, for any drug development project (including ASOs), the safety profile needs to be evaluated regardless of the presence of molecular off-target effects.

8) Ironically, it would be difficult to train a model like this with a drug which doesn't have unspecific effects, because you would not have enough data to train the model. This limits the potential applicability of this approach for drugs with better specificity and appears to be a major limitation of the current work.

We thank the reviewer for this comment, and agree, however we do not feel that this is a major limitation of this work. We selected BPN-15477 for this study for this very reason. The current work has yielded 214 mutations that cause human disease to be potential targets for this class of compounds, which was our goal. Further, we have validated several of our targets using EMGs and have now added data to show that even a small increase in splicing can translate into a clinically relevant increase in functional protein. Targeted chemical optimization of this class of compounds could lead to new therapies for many currently untreatable human genetic diseases.

Reviewer #3

*In this manuscript, authors present the development of a compound that corrects splicing of exon 20 of ELP1 gene carrying the major IVS20+6T>C mutation which is responsible for familial dysautonomia. Further, they applied machine learning approaches to evaluate the therapeutic potential of this compound to correct splicing in other human genetic diseases. **This manuscript is a good example of how deep learning techniques can identify sequence signatures and predict response to pharmacological modulation.** I am not an expert on machine learning so I will circumscribe my review comments to the aspects related with splicing.*

I assume that other reviewer (s) will comment on the machine learning aspects of the manuscript.

1) Figs. 4c-f: really the changes observed in the mutated lines with the BPN-15477 compound are very small (3-10% change). How much of that little change is caused by the compound or any other secondary effect?

Although we recognize that for some targets the splicing changes are modest, they are quite specific. The addition of our new data showing an increase of full-length protein after the treatment (for ELP1, LIPA and CFTR), as well as a specific increase in CFTR calcium channel function following treatment, further validate that the observed transcript changes are not the result of a secondary effect. Please also see the response to Reviewer 2, question 3.

2) Related with the previous point: how significant is the change in response to BPN-15477 in terms of physiologic relevance? I think this is something important to have this manuscript published in Nature Communications and it will for sure make the paper much stronger combining: splicing correction with a small molecule + physiological impact of that correction + machine learning.

This has been fully addressed above in response to Reviewer 2, question 3.

3) Please, indicate the number of independent experiments in each figure legend. It should be at least three.

We have now included in each figure legend the number of independent experimental replicates, and every experiment was repeated at least three times.

Reviewer #4

*Gao et al., present a deep-learning approach to predict splicing events that will be affected by the splicing modulator compound, BPN-15477, a more potent and efficacious derivative of the natural product, kinetin..... With splicing-related defects being a prevalent cause of disease and as more splicing modulator compounds are being discovered, **tools such as the one presented in this manuscript are valuable for understanding and predicting potentially harmful off-target effects as well as for identifying additional therapeutic targets for the molecules.....The study is a good proof-of-concept that presents a potentially valuable approach for identifying drug targets for this and other splice modulating compounds.***

There are a number of issues with the reporting in the manuscript (both minor and major) that should be addressed as detailed below.

1) Figure 1: b) should include EC50 data for both BPN-15477 and Kinetin. C) splicing correction is shown for the purpose of validating splicing correction by RT-PCR. The low end of the dose curve is missing as is the comparison to kinetin.

We have now included the EC50 data for both BPN-15477 and Kinetin in Figure 1C.

2) For the RNA-seq samples, were the cells treated a single time with the compound and then incubated for seven days? What is known about the stability of the compound in the

culture media? Why was this long incubation time selected? Fig 1b and c show results from fibroblast cells treated for only 24 hours.

The medium and drug were changed after three days of treatment. We have added more details regarding the treatment protocol for the fibroblast lines used in the RNA-seq experiment in the Material and Methods section of our revised manuscript. Our extensive experience in treating cell lines with BPN-15477 has shown that this compound is quite stable in the culture media. We performed several time-course experiments and we know that, once the compound is removed from the medium, mRNAs are back to baseline levels in 24 hours. Therefore, the fact that we observed splicing changes in treated cells when compared to DMSO, even after a long incubation period, demonstrates that the compound is still active.

3) Provide citation for statement: "This is exciting since a small increase in functional CFTR protein is associated with a significant improvement in patient phenotype".

We have added an expanded discussion of this topic (please see the response to Reviewer 2, Question 3) and have added references to the manuscript, as requested.

4) Provide raw data for all data points in Fig. 2C. This could be in the form of a Table that lists the name of the genes, the exon # that is changing and the PSI values.

We have now included a table that lists the name of the genes, the exon that is included/excluded and the PSI values for each data point in Fig. 2c as Supplementary Table 3.

5) In terms of selectivity, as presented on page 5 line 116, it would be informative to also include the number of alternatively spliced exons (triplets with two isoforms) that do not change. This number, I presume is the same as the unchanged-response set (382) exon triplets with two expressed isoforms described on p. 6 line 129.

This is correct, the number of alternatively spliced exons that do not change is 382, the same as the unchanged-response set.

6) In the bar graphs for the drug response plotting standardized probability, the values for the y-axis are not completely labeled with the bars extending beyond the y-axis labels, making it difficult to estimate the actual value.

Thank you, we have corrected this on all plots.

7) Fig. 3 and 4. These motifs are very short and degenerate, it is surprising that they could have any predictive value as they will be expected to be very frequent. Is there a way to score the motif match for each of these?

Although these motifs are quite short, please note their affects are not independent. Instead, the model uses a synergistic outcome of finding these motifs by adding up their convolved weights (Supplementary Fig. 2a), that is why the model is called convolutional neural network (CNN). For example, both Motif 18 and 25 had dominating AAG at the first three positions. The model "added up" the weight of them, which amplified the contribution of AAG, and pass it to the next stage of calculation towards final prediction. We regret not making this clear and we now updated the text in the manuscript to reflect this point. The

heatmap in Fig. 3b and Supplementary Fig. 4 actually measured the match of each motif at each position along the sequences that was important to make accurate predictions.

8) *The figure legend of Supp. Fig 3 indicates that the RT-PCR was performed in triplicate to make the bar plot. This use of technical replicates is not as robust as the use of experimental replicates and lessens the strength of the results. Similarly in Fig. 3 d-e, the legend states that each RT-PCR was done in duplicate and independently repeated three times for each minigene, does this mean that the same RNA sample from a single transfection/treatment was analyzed six times or was the transfection/treatment performed three times? Please clarify.*

The transfection/treatment was performed three independent times, as was the RT-PCR. As stated above we changed the figure legends to better clarify experimental and technical replicates.

9) *Fig. 4c,d it would be helpful to indicate the position of the mutation and the effect on the motif.*

We marked all mutated positions by coordinates relative to the 5' splice sites in red beneath the LOGO plots. We now mention this in the figure legends.

10) *The discussion could be expanded to consider in more detail the meaning of the results and how they relate to previous studies and the future of the molecule as a therapeutic. For example, the authors should comment on whether the motifs represent direct targets of the compound or just identify features of “weak” or “strong” splice sites that might be sensitive to subtle alterations in splicing in general induced by the compound. Further, it is a big leap to conclude from the motif analysis associated with sensitive exon 5'ss that the result is consistent with this class of compounds promoting the recruitment of U1 snRNP to non-canonical 5' splice sites. They cite a study by Yoshida et al., which shows that a kinetin-like molecule promotes U1-binding to the weak 5'ss of ELP1. However, Yoshida et al., actually go on to identify cis-acting sequence elements common to splicing events responsive to compound treatment and conclude that sequence elements that appear to act as hnRNP H binding exonic splicing silencers and may be the target of the compounds and that inactivating or interfering with these ESSs may allow more effective U1 snRNP binding simply because splicing is derepressed by the ESS.*

Thank you, we have modified the discussion slightly, however, we would like to point out that the main objective of the current work was to evaluate the effect of BPN-15477, a more potent analog of kinetin, on splicing and to use it to build and validate our machine learning model to predict new potential targets for therapy. It was not to evaluate the mechanism of action of this particular compound. It is likely that this class of SMCs are promoting recruitment of U1 to the splice site, either directly or indirectly. Further, as we noted above in response to Reviewer 2, further optimization would be needed to improve potency and specificity for each target. Nonetheless, our current work has yielded 214 mutations that cause human disease as potential targets for this class of compounds, which was our goal.

11) *Regarding Yoshida et al. further, the authors should discuss these previous results in more detail, perhaps commenting on the similarities or differences in identified motifs as well as the identified responsive splicing events.*

Although we recognize that BPN-15477 has a similar chemical structure to the kinetin derivative RECTAS, the goal of our study is completely different and does not lend itself to a direct comparison to the Yoshida study. One of the major differences between the two analyses is that while Yoshida et al. performed exon array analysis in treated FD fibroblasts we performed a deep RNA-seq analysis in treated WT fibroblasts. Treating FD cells with a splicing modifier increases the expression of ELP1 significantly, making it difficult to distinguish whether the observed changes in gene expression/splicing are due to the treatment itself or due to the increase in ELP1. Another major difference is that while Yoshida et al. has investigated differential exon expression, our study focuses on differential exon splicing using the inclusion ratio as gleaned from RNA-seq analysis. Exon expression can be influenced by gene expression while the exon inclusion ratio has been normalized for gene expression. It is not surprising, therefore, that we see differences. The novelty of our study lies in the use of a deep learning approach to identify new targets for a class of splicing modulator compounds.

Reviewers' Comments:

Reviewer #2:

Remarks to the Author:

Overall the response of the authors to the initial review comments appears satisfactory and they have made a fair effort to improve the manuscript.

- Effects of BPN-15477 in validation experiments are marginal and not very impressive.

While the comments in response to this criticism are broadly satisfactory and some examples of extremely low protein restoration levels mitigating disease, the evidence is overall marginal. In the case of Duchenne for instance, despite FDA approvals, no oligonucleotide or small molecule drug that restores dystrophin has yet been unequivocally shown, in appropriately controlled studies, to modify disease progression. Consensus in the field is that approx. 5% dystrophin is inadequate and non-therapeutic. While the inclusion of new data demonstrating that BPN-15477 treatment leads to ELP1 splicing correction and increases in ELP1 protein is welcome, the critical question is the effect of low level protein restoration on disease phenotype/progression, not necessarily on the ability to generate low levels of a deficient protein. The authors should comment critically on this.

- The authors identify specific motifs which explain ~95% of their model performance. It is essential that these motifs are independently validated by an orthogonal method e.g. an ASO which specifically targets those regions.

The authors have failed to respond adequately to this criticism. Validating this potential new mode of analysis and drug discovery should be accompanied by an attempt to independently validate these findings and target motifs using an independent/orthogonal approach. The authors have failed to answer this point.

- The paper draws parallels to ASO targetable exon-skipping diseases. However, the biggest drawback of BPN-15477 is the observation of unspecific splicing modulation (934 exon triples show significant modulation of the middle exon). The effects of this unspecific splicing alteration are not discussed but are certainly of concern from a therapeutic aspect. A more detailed analysis (ideally in vivo) and discussion of the limitations of the current approach are required.

Response to this point are broadly satisfactory. However the authors specific response 'We understand that specificity is an important consideration and we recognize that our compound needs additional medicinal chemistry optimization for any specific indication in order to increase the potency and decrease the off-target effects. Also, whether off-target effects observed in cells in a dish will lead to any undesired consequences to an organism is unclear' should be clearly reflected in the discussion i.e. that the existing compound requires further med chem and therefore may fail in drug development and also that the off-target effects observed in vitro may or may not have undesirable effects in a patient, again leading to failed drug development. The point is that lack of specificity, even at this low level, could result in failed drug development and/or adverse affects for patients and this shouldn't be dismissed.

- Finally, regarding competing financial interests. In addition to Dr Slaughaupt other authors on the manuscript (e.g. Johnson, Paquette) should be disclosed as having interests with respect to International Application Number PCT/US2016/013553

Reviewer #4:

Remarks to the Author:

The authors have improved the manuscript considerably by adding additional data and expanding on their descriptions and analysis. There are some critical points that need to be addressed in order to establish this approach as a robust means to identify relevant drug targets for disease therapeutics.

The experiments in Flp-In-293 cells of CFTR c.2988G>A were done with 60 uM BPN-15477 for 5 days with moderate splicing increase (Fig. 4C) and more dramatic protein increase Fig. 5b). The analysis of chloride channel function in treated CFBE cells were treated with only up to 10 uM drug for 3 days. Nonetheless, there was an increase in current suggesting an increase in CFTR protein. However, the authors do not correlate this increase in current with and increase in protein abundance or splicing correction, which can be easily done by collecting the cells after the functional assay is complete. This information is necessary to satisfactorily complete this very important experiment demonstrating the functional relevance of BPN-15477 treatment.

Figure 4d: There is a doublet in the BPN-15477-treated samples near the indicated inclusion. It looks like cryptic splicing has been activated. All amplicons should be sequenced to verify correct splicing and to identify any cryptic splicing.

In figure legends (2,3,4), please indicate explicitly the N used to calculate statistics. I assume that in Figure 4 the duplicates were averaged and that average was used as the independent repeat so N=3? My previous review asked this same question and it has not been satisfactorily addressed.

Reviewer #5:

Remarks to the Author:

The authors have satisfactorily addressed most of the comments from Reviewer 1. However, a few issues still exist:

1. Deep learning is prone to overfitting, especially when applied to small datasets. Therefore, I am quite surprised to see the high AUC value of the proposed CNN model, when trained on only a few hundred exons in each category. Could the authors clarify if the training and testing sets are completely separate from each other? That seems like the case, based on the text in the Methods section – but I think it'll be helpful to state this explicitly in the Results section too. Additionally, can the authors provide additional discussions and potential mechanistic explanations for why the CNN model appears to perform so well, when trained on such small training data?

2. As related to point #1 above, it would be useful to know whether this strategy can be similarly extended to other splicing-modulating small molecule compounds. RNA-seq data for other such compounds do exist, such as Risdisplam, and CLK small molecule inhibitor (<https://www.nature.com/articles/s41467-016-0008-7>). A demonstration on the generalizability of the proposed CNN model would substantially enhance its potential impact.

3. The authors evaluated the performance of the proposed CNN model using the AUC value, and the unchanged exon set contains only 382 exons. I agree the authors have a reasonable justification of using high-confidence unchanged exons for training the CNN. However, the use of a very small "negative" set makes the AUC value somewhat uninformative. The authors pointed out that the effect of BPN-15477 on splicing is highly selective (only 0.58% of expressed exon triplets are differentially spliced upon drug treatment). Therefore, in a real-life usage scenario, a predictive model with a high AUC value could end up having a low positive predictive value and high false discovery rate. This issue needs to be addressed and discussed in the manuscript, and it would be better to evaluate the performance of the model using the AUPR value.

4. The authors used RNA-seq data to examine drug induced exon splicing and intron retention events. The inclusion response exon set is characterized by weak 5' splice sites, consistent with the proposed MOA of this compound. Given this proposed MOA, and the compound's effect on exons with weak 5' splice sites, I think the authors should also analyze alternative 5' or 3' splice site events, to see if the compound preferentially alters 5' splice site usage leading to alternative 5' splice site events. This could be easily done by running one of the popular splicing tools (rMATS, SUPPA2, etc) on the RNA-seq data.

5. It is interesting to see that the majority of the drug-responsive exon splicing events fall into the exclusion response category, and these exons have "normal" splice site strength. Is it possible that BPN-15477 acts on these exons through an indirect mechanism, by perturbing a master splicing regulator? The authors noted that only 3.24% of genes were differentially expressed in BPN-15477 treated cells. Are there splicing regulators in these differentially expressed genes? Are there splicing regulators differentially spliced in BPN-15477 treated cells?

Reviewer #2 (Remarks to the Author):

Overall the response of the authors to the initial review comments **appears satisfactory** and they **have made a fair effort to improve the manuscript**.

- 1) *Effects of BPN-15477 in validation experiments are marginal and not very impressive. While the comments in response to this criticism are broadly satisfactory and some examples of extremely low protein restoration levels mitigating disease, the evidence is overall marginal. In the case of Duchenne for instance, despite FDA approvals, no oligonucleotide or small molecule drug that restores dystrophin has yet been unequivocally shown, in appropriately controlled studies, to modify disease progression. Consensus in the field is that approx. 5% dystrophin is inadequate and non-therapeutic. While the inclusion of new data demonstrating that BPN-15477 treatment leads to ELP1 splicing correction and increases in ELP1 protein is welcome, the critical question is the effect of low level protein restoration on disease phenotype/progression, not necessarily on the ability to generate low levels of a deficient protein. The authors should comment critically on this.*

While we agree with the reviewer that this is a critical question, it is one that can only be answered through clinical studies in humans and it no way invalidates the potential significance of our study.

While 5% dystrophin might be inadequate and non-therapeutic in DMD, this is not true for other diseases. Several studies have shown that CF protein levels as low as 10% of WT result in improved lung function and survival, and protein levels as low as 3% of WT can rescue pancreatic insufficiency, even in moderate to advanced stages of disease (McCague et al. Am J Respir Crit Care Med 2019; Ramsey et al. N Engl J Med 2011; Wainwright et al. N Engl J Med 2015; Taylor-Cousar et al. N Engl J Med 2017; Davies JC et al. N Engl J Med 2018; Keating et al. N Engl J Med 2018). Similarly, a 3% increase in lysosomal acid lipase (LAL or LIPA) residual enzyme activity is enough to distinguish Wolman disease, which is lethal in infancy, from the much milder CESD, therefore a 10% increase in functional LAL would be predicted to have clinical benefit (Aslanidis et al. Genomics 1996). In this manuscript we show that splicing correction after BPN-15477 treatment leads to increased protein production in disease relevant cellular models of LIPA (10% increase) and CFTR (20% increase). While these are representative confirmations, they are certainly over the predicted therapeutic threshold.

The power of our study is in predicting potential gene targets that might be amenable to a treatment strategy that targets splicing. In response to the initial review, we added significant confirmatory data in animal models (FD) patient cell lines (LIPA) and functional data in CF. The fact that we see functional protein increase after treatment is exciting, and further optimization for each disease is warranted. We in no way argue in this paper that the protein increase that we see will be therapeutic, that is not the point as we are using a small molecule that has not been optimized for the specific targets yet. We are arguing that we have identified 155 gene targets that harbor splice mutations that cause human disease (in some cases previously unrecognized) using a deep learning model and we have functionally validated several of our hits, setting the stage for future studies aimed at developing and testing therapeutics in human clinical trials.

- 2) *The authors identify specific motifs which explain ~95% of their model performance. It is essential that these motifs are independently validated by an orthogonal method e.g. an ASO which specifically targets those regions.*

The authors have failed to respond adequately to this criticism. Validating this potential new mode of analysis and drug discovery should be accompanied by an attempt to independently validate these findings and target motifs using an independent/orthogonal approach. The authors have failed to answer this point.

Respectively, we do not agree that independent validation of specific motifs using an ASO is at all relevant to the validity of our results. As we state in the paper and inherent in the deep learning approach, the treatment response *is not determined by any of these motifs independently*, which is the strength of using a modeling approach. Therefore, the inactivation of one specific motif using an ASO would not necessarily result in a failure of treatment response, again highlighting the complexity and the power of the model.

Importantly, several of our predictions were independently validated by treating disease relevant cellular models and confirming splice correction and functional protein increase, which is ultimately a more powerful approach for validation since the point of our study was to identify gene targets harboring disease causing mutations predicted to respond to a specific small molecule. We do not suggest that this is a new method for drug discovery, but rather a potential method to further interrogate existing compounds for their effect on a transcriptome level.

- 3) *The paper draws parallels to ASO targetable exon-skipping diseases. However, the biggest drawback of BPN-15477 is the observation of unspecific splicing modulation (934 exon triples show significant modulation of the middle exon). The effects of this unspecific splicing alteration are not discussed but are certainly of concern from a therapeutic aspect. A more detailed analysis (ideally in vivo) and discussion of the limitations of the current approach are required.*

Response to this point are broadly satisfactory. However the authors specific response ‘We understand that specificity is an important consideration and we recognize that our compound needs additional medicinal chemistry optimization for any specific indication in order to increase the potency and decrease the off-target effects. Also, whether off-target effects observed in cells in a dish will lead to any undesired consequences to an organism is unclear’ should be clearly reflected in the discussion i.e. that the existing compound requires further med chem and therefore may fail in drug development and also that the off-target effects observed in vitro may or may not have undesirable effects in a patient, again leading to failed drug development. The point is that lack of specificity, even at this low level, could result in failed drug development and/or adverse affects for patients and this shouldn’t be dismissed.

We are pleased that the reviewer found our response to this point satisfactory and as suggested by the reviewer we have clarified this point in the discussion. It is our view that the more potential therapies that exist for a specific disease, the better. It may be that small molecules will be complementary to ASO therapies, or vice versa. It is too soon to predict if one therapeutic mechanism will be better than another.

- 4) *Finally, regarding competing financial interests. In addition to Dr Slaugenhaupt other authors on the manuscript (e.g. Johnson, Paquette) should be disclosed as having interests with respect to International Application Number PCT/US2016/013553*

We have edited the Competing Financial Interests section and include Drs. Johnson and Paquette as inventors in the International Application Number PCT/US2016/013553. However, they have no financial interests as they have assigned to MGH as required by the NIH Blueprint Therapeutics Network agreements.

Reviewer #4 (Remarks to the Author):

*The authors **have improved the manuscript considerably by adding additional data and expanding on their descriptions and analysis.** There are some critical points that need to be addressed in order to establish this approach as a robust means to identify relevant drug targets for disease therapeutics.*

- 1) *The experiments in Flp-In-293 cells of CFTR c.2988G>A were done with 60 uM BPN-15477 for 5 days with moderate splicing increase (Fig. 4C) and more dramatic protein increase Fig. 5b). The analysis of chloride channel function in treated CFBE cells were treated with only up to 10 uM drug for 3 days. Nonetheless, there was an increase in current suggesting an increase in CFTR protein. However, the authors do not correlate this increase in current with and increase in protein abundance or splicing correction, which can be easily done by collecting the cells after the functional assay is complete. This information is necessary to satisfactorily complete this very important experiment demonstrating the functional relevance of BPN-15477 treatment.*

We have now included CFTR splicing analysis in treated CFBE cells and show that the improvement in chloride channel function correlates with a significant increase in exon 18 inclusion (Supplementary Figure 6).

- 2) *Figure 4d: There is a doublet in the BPN-15477-treated samples near the indicated inclusion. It looks like cryptic splicing has been activated. All amplicons should be sequenced to verify correct splicing and to identify any cryptic splicing.*

As correctly stated by Reviewer 4, in the mutant MLH1 minigene we observed activation of a cryptic 5' splice site at +31. The identity of each band was verified by sequence analysis. In the legend of Figure 4 we have now included this information.

- 3) *In figure legends (2,3,4), please indicate explicitly the N used to calculate statistics. I assume that in Figure 4 the duplicates were averaged and that average was used as the independent repeat so N=3? My previous review asked this same question and it has not been satisfactorily addressed.*

We have now included the N in each figure legend. We apologize that this was not clear in our original response. Since we did not average the duplicates and we considered every single value independently the N is 6.

Reviewer #5 (Remarks to the Author):

*The authors **have satisfactorily addressed most of the comments from Reviewer 1.** However, a few issues still exist:*

Major comments:

- 1) *Deep learning is prone to overfitting, especially when applied to small datasets. Therefore, I am quite surprised to see the high AUC value of the proposed CNN model, when trained on only a few hundred exons in each category. Could the authors clarify if the training and testing sets are completely separate from each other? That seems like the case, based on the text in the Methods section – but I think it'll be helpful to state this explicitly in the Results section too. Additionally, can the authors provide additional discussions and potential mechanistic explanations for why the CNN model appears to perform so well, when trained on such small training data?*

As suggested by the reviewer, we now state in the Results section (Line 150-152) that the training and testing data are separate from each other. We've also included a new discussion point (Line 346-357) suggesting the potential reasons why our CNN's performance is robust, including 1) the trained model highly scored a short region of ~8bp (out of a 400bp training region) flanking the 5' ss of the middle exon, while the scores of other regions were almost zero (Supplementary Fig. 4). This suggests that the treatment effect might be in a condensed region relative to the 5' ss. 2) We empirically avoided overfitting by applying the L1-regularization (coefficient = 0.6) in the convolutional layers and the dropout strategy (see Methods) in the hidden layer during model training. Both of these factors made the model converge quickly (in twelve epochs, Supplementary Fig. 2b) on a small sample size.

- 2) *As related to point #1 above, it would be useful to know whether this strategy can be similarly extended to other splicing-modulating small molecule compounds. RNA-seq data for other such compounds do exist, such as Risdisplam, and CLK small molecule inhibitor (<https://www.nature.com/articles/s41467-016-0008-7>). A demonstration on the generalizability of the proposed CNN model would substantially enhance its potential impact.*

We appreciate the Reviewer's suggestion to demonstrate the generalizability of the model and we agree this will be a critical area of exploration. Indeed, we have written an entire grant proposal to partially pursue these questions, as well to include other molecules derived from kinetin in order to identify additional gene targets. However, these additional experiments and analyses are well beyond the scope of this paper, where we focused on creating the model, identifying gene targets for BPN-15477, and validating our predictions in relevant cellular patient models.

- 3) *The authors evaluated the performance of the proposed CNN model using the AUC value, and the unchanged exon set contains only 382 exons. I agree the authors have a reasonable justification of using high-confidence unchanged exons for training the CNN. However, the use of a very small "negative" set makes the AUC value somewhat uninformative. The authors pointed out that the effect of BPN-15477 on splicing is highly selective (only 0.58% of expressed exon triplets are differentially spliced upon drug treatment). Therefore, in a real-life usage scenario, a predictive model with a high AUC value could end up having a low positive predictive value and high false discovery rate. This issue needs to be addressed and discussed in the manuscript, and it would be better to evaluate the performance of the model using the AUPR value.*

Thank you for this helpful comment. As suggested, we have re-estimated the model performance using AUPR and we found that the prediction for inclusion, exclusion, and unchanged sequences in the test set achieved a PR-AUC of 0.8, 0.84 and 0.61 respectively, which is consistent with our initial observations using AUC (Supplementary Fig. 2c *right panel*). This additional analysis proves the robustness of the model and confirms that a higher false discovery rate is unlikely. The new figure for AUPR estimation is shown below and has been added to Supplementary Fig. 2c right panel.

We agree with the Reviewer that the effect of BPN-15477 on the WT transcriptome is highly selective with only 0.58% of exon triplets responding to the drug. This is a reflection of the efficiency of the splicing machinery in WT cells where most middle exons are included 100% of the time. However, our model is particularly valuable when a mutation leads to a splicing error. Given that the effect of a particular mutation on splicing is typically not annotated and highly underestimated, our model provides an approach to facilitate treatment development for diseases caused by splice mutations. We have now noted this in the Discussion (Line 366-369).

- 4) *The authors used RNA-seq data to examine drug induced exon splicing and intron retention events. The inclusion response exon set is characterized by weak 5' splice sites, consistent with the proposed MOA of this compound. Given this proposed MOA, and the compound's effect on exons with weak 5' splice sites, I think the authors should also analyze alternative 5' or 3' splice site events, to see if the compound preferentially alters 5' splice site usage leading to alternative 5' splice site events. This could be easily done by running one of the popular splicing tools (rMATS, SUPPA2, etc) on the RNA-seq data.*

As suggested by the Reviewer, this additional analysis will not change the results of our study since we focused on using alternative exon inclusion/exclusion as evaluated by exon triplets. In other words, examining alternative 5' and 3' ss usage is merely a different way of viewing two unique exon triplets that would differ by the length of the middle exon. Given this, we did not add this analysis to the supplement since these events would have already been identified. Nonetheless, we agree that it is an interesting question and in order to answer the Reviewer, we applied *rMATS* on our RNASeq data with the option "--paired-stats" for using a paired statistical model. We defined alternative 5' ss or 3' ss with FDRs < 0.1 and absolute changes of splicing ≥ 0.1 , which is as the same magnitude of change we used to define differential exon inclusion and exclusion in the paper. By doing so, we found 132 (3.06%, out of 4,312) alternative 5' ss events and 75 (1.11%, out of 6,742) alternative 3' ss events (figures shown below). As expected, these events overlap significantly with our differential exon inclusion analysis.

- 5) *It is interesting to see that the majority of the drug-responsive exon splicing events fall into the exclusion response category, and these exons have “normal” splice site strength. Is it possible that BPN-15477 acts on these exons through an indirect mechanism, by perturbing a master splicing regulator? The authors noted that only 3.24% of genes were differentially expressed in BPN-15477 treated cells. Are there splicing regulators in these differentially expressed genes? Are there splicing regulators differentially spliced in BPN-15477 treated cells?*

This is an interesting question and would require a detailed study of the precise mechanism of action of this compound. We did, however, examine the functional annotations for the differentially expressed genes (DEGs) and differentially spliced exon-triplet (DSEs). There are 157 (0.88% out of 17,812 expressed genes) splicing regulators in Gene Ontology database (GO:0043484). Among 577 DEGs, there were five (0.87%) splicing regulators: *LPHN2*, *NEO1*, *NGR1*, *SLC22A17* and *PILRB*. The first four were upregulated while the last one is downregulated.

Among the 934 DSEs responsive to BPN-15477, thirteen (or 1.40%) were located in eight annotated splicing regulators: *CLK1*, *FAM172A*, *FXR2*, *HNRNPA2B1*, *MBNL1*, *SETX*, *SLC38A2*, *SNRNP70* and *TIA1*. Twelve of these exon triplets had increased middle exon retention after BPN-15477 treatment.

While the above results suggest that BPN-15477 did not alter either splicing or expression of a known annotated splicing regulator more often than random, it is impossible to know for sure what the downstream consequences of these changes are, and we are unable to study this directly using RNAseq data. As mentioned in the response above, our results set the stage for a future study, perhaps using RNA Immunoprecipitation Chip (RIP) assays, that will uncover the precise mechanism of action of this drug.

Reviewers' Comments:

Reviewer #2:

Remarks to the Author:

The manuscript is improved and all comments have been addressed satisfactorily except for that regarding the lack of independent biological validation of the treatment responsive sequence signatures identified. In my opinion this continues to represent a weakness in the manuscript.

Reviewer #4:

Remarks to the Author:

I appreciate the authors' thoughtful and largely corrective responses to my previous concerns. I have only two remaining critiques that should be addressed.

Figure 4: If the experiment was done three times and each sample from each experiment was analyzed twice, the n used for statistics and graphed should be 3 not 6 (the experiment was replicated three times, not six times). By using n=6, the authors are artificially lowering the p-value and increasing the power of the result by representing each sample twice.

Supplementary Fig. 6: The lanes in the gel should be labeled. I assume that the first three lanes are untreated and the last three lanes are treated with BPN-15477? Was the untreated actually vehicle-treated (which would be the appropriate control since DMSO is known to have effects on splicing)?

Reviewer #5:

Remarks to the Author:

The authors have satisfactorily addressed most of my comments from the previous round of review. I only have one remaining issue. The authors' response to my comment #3 regarding the performance of the CNN model seems to miss the point. As indicated by the data and noted by the authors, the effect of BPN-15477 on splicing is highly selective (only 0.58% of expressed exon triplets are differentially spliced upon drug treatment). Predicting drug-responsive exons in this situation is similar to diagnosing a disease with a very low prevalence in the population – a test with a low false positive rate could still end up having a high false discovery rate because the prevalence is low. In this work, the authors used a very small "negative" (i.e. unchanged) exon set, that contains only 382 exons, for training and evaluating the CNN. In reality, the size of the negative exon set is far larger than the size of the positive exon set. Therefore, regardless of whether AUC or AUPR is used for performance evaluation, the use of this small negative exon set would make the performance inflated and unrealistic. One way to assess this issue is to apply the CNN model to the whole transcriptome, and at various cutoffs for the prediction score, use the RNA-seq data (and/or RT-PCR experiments) to assess what percentage of positive predictions are indeed correct. This would provide a more realistic estimation of the false discovery rate. This can be done on all expressed exon triplets, or on a subset of expressed exon triplets with weak splice sites for the middle exon (given that the drug seems to particularly impact exons with weak splice sites, as noted by the authors).

I should note that regardless of the results of the analysis proposed above, I don't view this issue a deal breaker. Nonetheless, I think it's important for the authors to present the performance and applicability of their model in a more realistic manner, that matches the real-life usage scenario of this model.

Reviewer #4 (Remarks to the Author):

*I appreciate the authors' thoughtful and **largely corrective responses to my previous concerns**. I have **only two remaining critiques** that should be addressed.*

- 1) *Figure 4: If the experiment was done three times and each sample from each experiment was analyzed twice, the n used for statistics and graphed should be 3 not 6 (the experiment was replicated three times, not six times). By using $n=6$, the authors are artificially lowering the p -value and increasing the power of the result by representing each sample twice.*

We regret the confusion regarding technical and biological replicates introduced with our usage of the word “duplicate”. We did not analyze the same samples twice. We performed two independent experiments and thus across these independent experiments and treatment groups we performed 6 replicates. We have corrected the legends accordingly.

- 2) *Supplementary Fig. 6: The lanes in the gel should be labeled. I assume that the first three lanes are untreated and the last three lanes are treated with BPN-15477? Was the untreated actually vehicle-treated (which would be the appropriate control since DMSO is known to have effects on splicing)?*

We now have labelled the lanes in Supplementary Figure 6. The first three lanes represent the *CFTR* splicing analysis in DMSO-treated cells while the last three lanes represent the *CFTR* splicing analysis in BPN-15477-treated cells.

Reviewer #5 (Remarks to the Author):

*The authors have **satisfactorily addressed most of my comments** from the previous round of review. I **only have one remaining issue**...I should note that **regardless of the results of the analysis proposed above, I don't view this issue a deal breaker**. Nonetheless, I think it's important for the authors to present the performance and applicability of their model in a more realistic manner, that matches the real-life usage scenario of this model.*

- 1) *In reality, the size of the negative exon set is far larger than the size of the positive exon set. Therefore, regardless of whether AUC or AUPR is used for performance evaluation, the use of this small negative exon set would make the performance inflated and unrealistic. One way to assess this issue is to apply the CNN model to the whole transcriptome, and at various cutoffs for the prediction score, use the RNA-seq data (and/or RT-PCR experiments) to assess what percentage of positive predictions are indeed correct...This can be done on all expressed exon triplets, or on a subset of expressed exon triplets with weak splice sites for the middle exon (given that the drug seems to particularly impact exons with weak splice sites, as noted by the authors).*

We did not initially conduct these analyses due to the inherent noise in a transcriptome-wide analysis, but we appreciate this perspective and have taken the Reviewer's guidance to evaluate the CNN model using all available triplets. We included all exon-triplets with PSI between 0.01 and 0.99 before or after treatment. We defined the “true inclusion response” group as consisting of triplets with PSI changes ≥ 0.01 after treatment, while the “true exclusion response” group consisted of triplets with PSI changes ≤ -0.01 after treatment. The remaining triplets were considered the “true unchanged” group. These groups contained 5.70%, 14.08% and 80.22% of the 18,766 exon triplets, respectively. Application

of our CNN model on these data yielded an ROC-AUC of 0.70, 0.67 and 0.67 for inclusion, exclusion, and unchanged predictions, respectively (see below). As expected and noted by the Reviewer, these analyses display reduced performance by comparison to the highly curated true positive triplets used in the test set, however they remain highly significant by comparison to a random classifier (see below). We have now included these comparisons in the methods and Supplementary Figure 2c.

Reviewers' Comments:

Reviewer #4:

Remarks to the Author:

Thank you for making the clarifications and corrections in your manuscript. I am very enthusiastic about the study.

Reviewer #5:

Remarks to the Author:

The authors have addressed my remaining concern. The paper is ready for publication.